DOI: 10.1038/s41467-018-05108-5　　**OPEN**

# Mapping the sensing spots of aerolysin for single oligonucleotides analysis

Chan Cao[1,2], Meng-Yin Li[1], Nuria Cirauqui[2,3], Ya-Qian Wang[1], Matteo Dal Peraro[2], He Tian ⬤ [1] & Yi-Tao Long[1]

Nanopore sensing is a powerful single-molecule method for DNA and protein sequencing. Recent studies have demonstrated that aerolysin exhibits a high sensitivity for single-molecule detection. However, the lack of the atomic resolution structure of aerolysin pore has hindered the understanding of its sensing capabilities. Herein, we integrate nanopore experimental results and molecular simulations based on a recent pore structural model to precisely map the sensing spots of this toxin for ssDNA translocation. Rationally probing ssDNA length and composition upon pore translocation provides new important insights for molecular determinants of the aerolysin nanopore. Computational and experimental results reveal two critical sensing spots (R220, K238) generating two constriction points along the pore lumen. Taking advantage of the sensing spots, all four nucleobases, cytosine methylation and oxidation of guanine can be clearly identified in a mixture sample. The results provide evidence for the potential of aerolysin as a nanosensor for DNA sequencing.

[1] Key Laboratory for Advanced Materials & School of Chemistry and Molecular Engineering, East China University of Science and Technology, Shanghai 200237, P.R. China. [2] Institute of Bioengineering, School of Life Sciences, Ecole Polytechnique Fédérale de Lausanne (EPFL), 1015 Lausanne, Switzerland. [3] Department of Pharmaceutical Biotechnology, Universidade Federal do Rio de Janeiro, 21941-902 Rio de Janeiro, Brazil. Correspondence and requests for materials should be addressed to M.D.P. (email: matteo.dalperaro@epfl.ch) or to Y.-T.L. (email: ytlong@ecust.edu.cn)

Nanopore sensing has developed into a powerful tool for single-molecule analysis, especially for single-molecule DNA sequencing in a rapid, low-cost, and label-free way without the need of PCR amplification[1]. Measuring the fluctuations of ion current as four kinds of individual bases of DNA (A, T, C, and G) transport through a nanopore allows the real-time sequencing of single-strand DNA (ssDNA)[2]. So far, some membrane proteins and porins have been explored as a biological nanopore for DNA analysis, such as α-hemolysin[3], MspA[4], phi29 DNA-packaging nanomotor[5], ClyA[6], FhuA[7], aerolysin[8], and CsgG[9,10]. Compared with the commonly used α-hemolysin, whose channel constriction is nearly 1.4 nm, MspA has higher sensitivity in current resolution, since its funnel-like aperture is only ~1.2 nm in diameter at the narrowest point, with a translocation height of less than 0.6 nm[11]. Therefore, the architecture of the protein pore, and especially pore constriction, plays a crucial role in nanopore sensing ability.

Aerolysin, produced by *Aeromonas sp.*, has been used as a biological nanopore to study the conformation and length of peptides[12,13], dynamics of unfolded proteins[14], enzyme activity[15,16], and mass of PEGs[17]. Compared to α-hemolysin and MspA, aerolysin exhibited a higher sensitivity in current and duration for oligonucleotides analysis[18]. Recently, aerolysin has been rationally designed to control the selectivity and sensitivity for oligonucleotide sensing via site-directed mutagenesis[19]. Aerolysin shows an optimal single-nucleobase resolution and translocation rate of 2.0 ms nt$^{-1}$ for oligonucleotides in wild-type conformation[18] without the need of any additional complement, such as pore engineering[20], DNA immobilization[21], adapter incorporation[22], or the use of enzymes[2]. Aerolysin could directly detect ssDNA as short as dinucleotides, and resolve individual short oligonucleotides that range from 2 to 10 bases in length with high sensitivity, achieving the real-time monitoring of the step-wise cleavage of oligonucleotides by Exo I[23]. In the future, taking advantage of the high spatial and temporal resolution of aerolysin, the operations of DNA analysis may be simplified and its accuracy could be significantly improved.

However, a high-resolution structure of the mature aerolysin pore is still missing to date, hindering a much deeper understanding of its capabilities for single-molecule analysis and the development of further applications. Nonetheless, aerolysin pore models have been recently proposed based on near-atomic resolution cryo-electron microscopy (EM) structures (3.9–4.5 Å) of its mutated prepore and quasi-pore states, along with a 7.9-Å resolution model of the wild-type pore state[24]. This structural analysis revealed a heptameric pore architecture featuring a very long (~10 nm) membrane spanning the β-barrel pore channel, which resembles the one found for anthrax toxin[25] in dimensions, and is much longer than the α-hemolysin pore[3]. Moreover, aerolysin presents a new fold constituted by two concentric β-barrels at the top of the transmembrane pore, held together mainly by hydrophobic interactions, which are suggested to be responsible for the prion-like ultra-stability of this toxin[24].

Herein, we combine the results of nanopore experiments along with molecular modeling and simulation to precisely map the sensing spots of this protein for oligonucleotides analysis. Our results revealed two clear sensing spots located at the two main constriction points across the pore lumen, located at amino-acid residues R220 and K238. We show that the first spot at position R220 is mainly responsible for specifically recognizing different nucleobases. Moreover, based on the construction features of the pore entry, we demonstrated the preference of DNA translocation entering from the 3′ end, consistently with other pore-forming toxins (PFTs) such as α-hemolysin[26,27]. Therefore, by rationally probing ssDNA length and composition upon pore translocation, we gained new important molecular insights that which could

help the future design of aerolysin variants with improved capabilities for sequencing.

## Results

**Aerolysin pore is fully blocked by a 14-base oligonucleotide.** To determine the blockade characteristics of the aerolysin nanopore, we performed single-channel experiments of polydeoxyadenines ($dA_n$) with different lengths, from $dA_{10}$ to $dA_{20}$. The experimental schematic diagram is shown in Fig. 1a, and it is constituted by a lipid membrane that separates the chamber into two parts, *cis* and *trans*, with the aerolysin pore connecting *trans* and *cis* side on the membrane. An applied transmembrane potential pulls the negatively charged ssDNA from the *cis* to the *trans* side. According to the typical current traces of $dA_n$ (Fig. 1b), the level of blockade current gradually closed to 0 pA as the length of oligonucleotide strands increased, corresponding to a full decrease of residual current and an increase in the duration of DNA translocation (Fig. 1c–e). We found that these current values presented a strong dependence with the length of oligonucleotides for $dA_n$ shorter than 14 bases ($n<14$), for which the value of $I/I_0$ decreased from $0.35 \pm 0.01$ to $0.05 \pm 0.01$ a.u. with $n=10–14$. (Here, $I_0$ represents the value of open pore current, while $I$ is the value of residual current; thus $I/I_0$ indicates the current ratio of blockade. Mean values (± s.d.) in at least five separate experiments). In contrast, when the length of $dA_n$ is no shorter than 14, the $I/I_0$ remained almost unchanged with a constant value of $0.07 \pm 0.02$ a.u. (mean values (± s.d.) in at least five separate experiments) (Fig. 1c and d). For all measured $dA_n$, $I/I_0$ followed a Gaussian distribution (Fig. 1d, inset), while the distribution of the translocation time was fitted by a falling exponential function (Fig. 1e, inset). Moreover, the statistical data at different applied voltages indicated that the translocation duration of each $dA_n$ significantly decreased with increased applied voltages (Supplementary Fig. 1), which demonstrated that $dA_n$ are mostly transported through the aerolysin nanopore. Notably, the duration of $dA_{14}$ is surprisingly longer compared to other tested oligonucleotides (Fig. 1e), suggesting that the interaction between $dA_{14}$ and the aerolysin nanopore is much stronger, likely due to the optimal spatial filling of $dA_{14}$ within the ~10 nm long aerolysin channel.

To prove this hypothesis and better characterize the main interactions between oligonucleotides and aerolysin, which may explain the longer duration time observed for $dA_{14}$, we performed 250 ns of molecular dynamics (MD) simulations of $dA_{14}$ fully inserted in the pore lumen. The initial conformation was obtained by steered MD, as explained in the methods section. In Fig. 1f (and in Supplementary Table 1), the main interactions between the $dA_{14}$ and aerolysin are summarized. Briefly, stable salt-bridge interactions are observed between the phosphate groups and the positive aerolysin amino acids, mainly with R282 and R220 at the pore entry, and K238 and K242 at the pore exit. Moreover, nucleobases are forming cation-π interactions with positively charged side chains, as well as hydrogen bonds and van der Waals contacts with several lumen residues (e.g., D216, N226, E254, E258, N262, and Q268). Altogether, these interactions are able to keep $dA_{14}$ stable in its extended conformation within the two main positively charged rings defined by R220 and R282 at the top of the channel and K238, K242, and K244 at its bottom. This observation agrees with previous MD studies using the MspA pore, which revealed that the inclusion of positive side chains in the lumen resulted in an increase in translocation time[28]. The optimal steric and electrostatic match of $dA_{14}$ within the lumen is likely at the origin of the observed increased duration time. However, by comparison, shorter ($dA_{10}$) and longer ($dA_{20}$) oligonucleotides did not show the same perfect match during MD

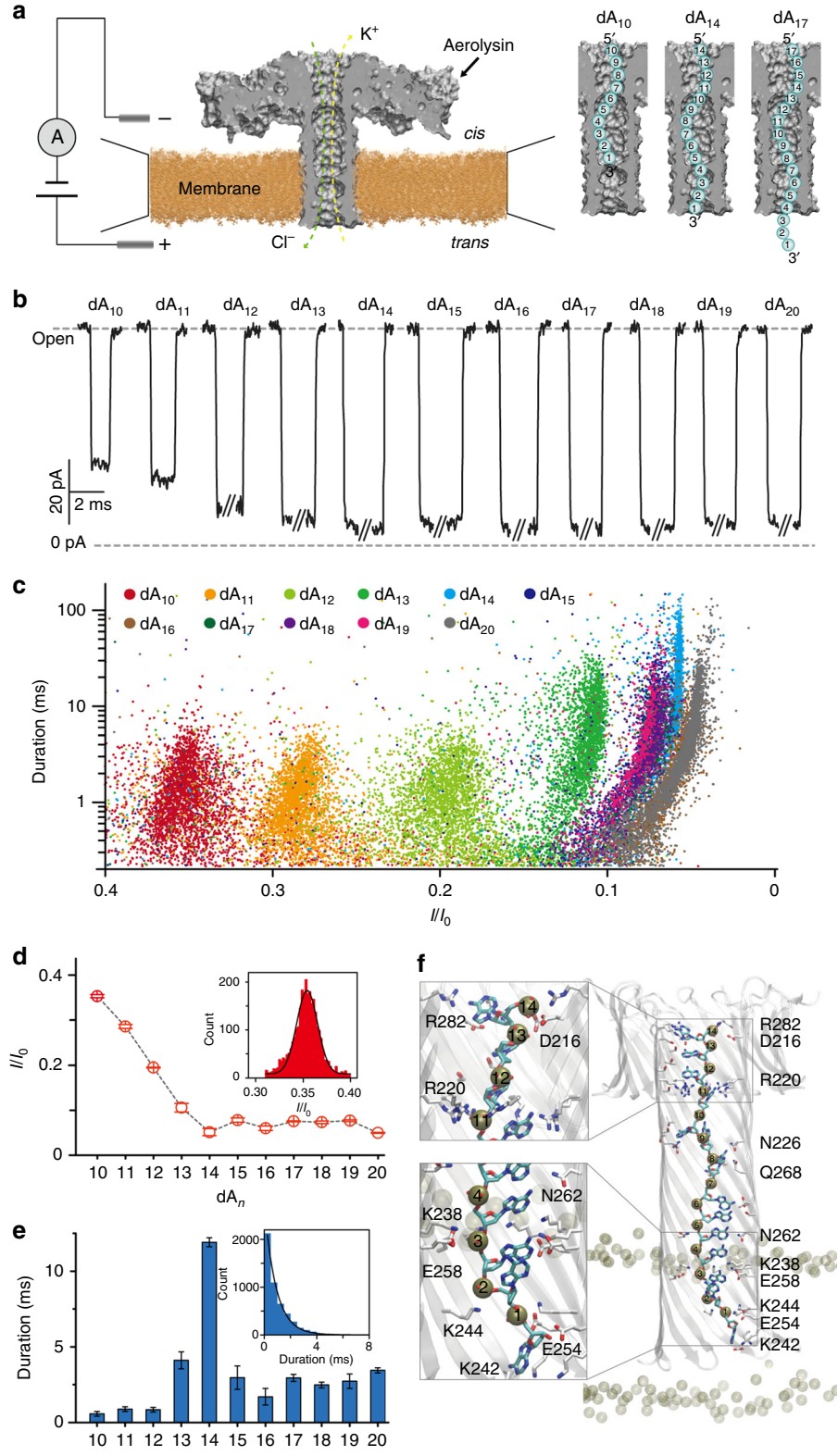

**Fig. 1** Single-channel recording of dA$_n$ ($n = 10$–20) with aerolysin nanopores. **a** Schematic representation of the aerolysin nanopore system, and dA$_n$ (cyan circles) of different lengths transported through it (only the pore lumen is shown for the sake of clarity). Typical current traces (**b**), scatter plots (**c**), statistical analysis of $I/I_0$ (**d**), and corresponding durations (**e**) caused by different lengths of dA$_n$ ($n = 10$–20) using aerolysin nanopores. All data were obtained in 1.0 M KCl, 10 mM Tris, and 1.0 mM EDTA, pH=8.0, 24 ± 2 °C at the bias potential of +140 mV. The errors of the data were based on at least five separate measurements. **f** MD snapshot of dA$_{14}$ fully inserted in the pore. The molecules are colored as follows: nitrogen blue, oxygen red, carbon cyan (for DNA) or white (for protein), and phosphate tan. The membrane and DNA phosphate atoms are shown as spheres. The dA$_{14}$ nucleotides are numbered according to the text. The insets show the most important interactions. The image was created with the Visualization Molecular Dynamics (VMD) software[29]

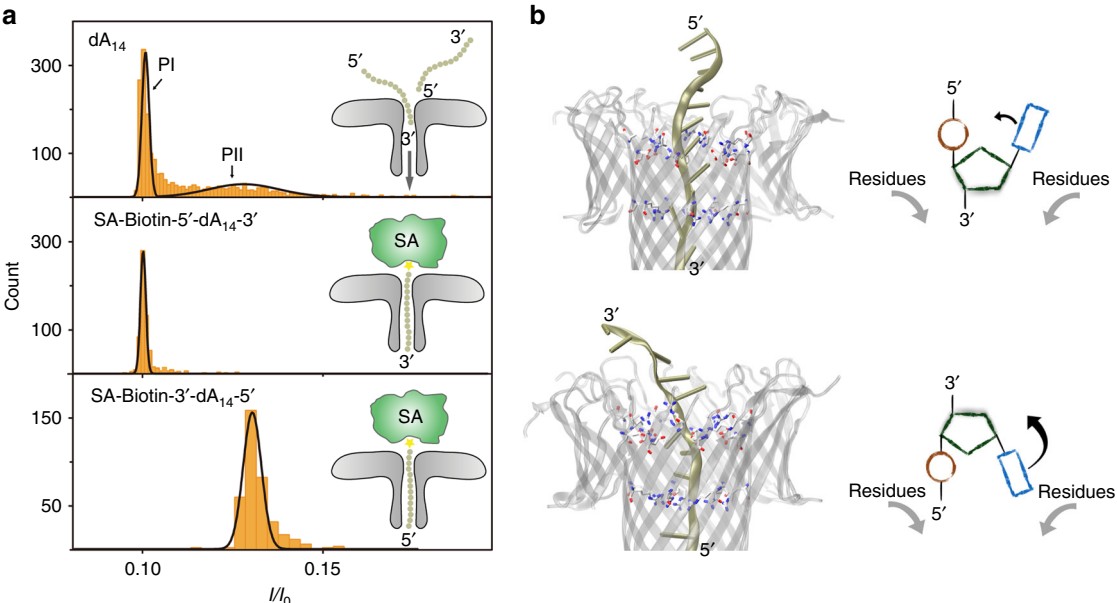

**Fig. 2** Analysis of oligonucleotide translocation direction through aerolysin nanopores. **a** Current blockade histograms with corresponding Gaussian fits (left) and schematic diagrams (right) of $dA_{14}$ (top), SA-Biotin-5'-$dA_{14}$-3' (middle), and SA-Biotin-3'-$dA_{14}$-5' (bottom) inside an aerolysin pore, respectively. All data were acquired at 24 ± 2 °C with the applied voltage of 100 mV in 3.0 M KCl, 10 mM Tris and 1.0 mM EDTA buffered at pH 8.0 in the presence of 5.0 μM oligonucleotide (translocation experiment) or 5.0 μM oligonucleotide with 1.0 μM streptavidin (immobilization experiment). **b** Steric hindrance imposed by the equatorial location of the base in the desoxi-D-ribose while it translocates directed by 5' (top) or 3' (bottom) end. Left, snapshots of SMD showing residues at the pore entry and the DNA represented by a tan ribbon. Right, scheme with the aerolysin amino acids represented by gray arrows, and the DNA phosphate, sugar and base colored red, green, and blue, respectively

simulations (Supplementary Fig. 2). For the longer $dA_{20}$, the results showed that this oligonucleotide remains in fact more extended than the pore length, resulting in the accumulation of the remaining bases at the pore entry or exit depending on starting conditions. On the other hand, $dA_{10}$ does not completely fill the pore in length, and the DNA tends to accumulate at the pore exit. Contrary to the optimal match of $dA_{14}$, shorter or longer oligonucleotides may accelerate their exit from the pore by facilitating a faster restoration of ion current thanks to conformations that allow for a partially filled pore lumen (e.g., $dA_{10}$) or solvated bases in the bulk (e.g., $dA_{20}$).

**ssDNA shows a preferred translocation direction by 3' end**. The scatter plots of Fig. 1c show a unique direction of entry for $dA_n$. In order to understand which this preferential direction was, we used a higher salt concentration (3.0 M), which produced two well-defined peaks in the $I/I_0$ histograms of $dA_{14}$, each corresponding to a different entry direction (Fig. 2a). One peak represents a lower blockade with a $I/I_0$ value of 0.10, which contains 79.4% of the total number of events (defined as PI hereafter), while the other (PII hereafter), containing the remaining 20.6% of events, presents a higher average $I/I_0$ value (0.13). These two peaks may represent two distinct translocation mechanisms, initiated either by the 5' or by the 3' end, similarly to that reported by previous studies of α-hemolysin[30]. To confirm this hypothesis and to further determine the assignment of these two peaks, a streptavidin-biotin system was designed to control the orientation of $dA_{14}$ while entering the pore. Streptavidin (SA) from *Streptomyces avidinii*[31] has a size of ~10 × 10 × 125 nm³, which prevents it from entering the aerolysin pore. The biotinylated $dA_{14}$ could specifically bind to SA and thus allow $dA_{14}$ to enter the pore only in one direction. To eliminate the influence of biotin for $I/I_0$, the translocations of Biotin-5'-$dA_{14}$-3' and Biotin-3'-$dA_{14}$-5' were examined, respectively (Supplementary Fig. 3). Unlike the $dA_{14}$, the $I/I_0$ histograms of both SA-Biotin-5'-$dA_{14}$-3'

and SA-Biotin 3'-$dA_{14}$-5' show only one peak, proving that SA immobilizes the free $dA_{14}$ inside the pore and ensures one certain orientation (Fig. 2a). Moreover, considering the biotin influence, the corrected $I/I_0$ value of SA-Biotin-5'-$dA_{14}$-3' ($I/I_0^{3'}$ = 0.10) is equal to $I/I_0$ of PI for $dA_{14}$, which means that the peak with more than 79% of total events was induced by the $dA_{14}$ entering the aerolysin pore from the 3' end, while the corrected $I/I_0$ value of SA-Biotin-3'-$dA_{14}$-5' ($I/I_0^{5'}$ = 0.13) is consistent with the PII of $dA_{14}$. Therefore, while single oligonucleotides are able to cross the aerolysin channel directed either by 3' or 5' end (similarly to other PFTs[26,30]), the $dA_n$ tested above show a significant preference for translocation via the 3' end.

In order to understand why DNA would more favorably translocate by 3' end, contrary to what was expected due to the extra negative charge present at the 5' end, we performed Steered MD (SMD) of the DNA, pulling it inside the pore either by its 3' or by its 5' end. We could observe a relatively higher force needed for translocation by the 5' end (Supplementary Fig. 4). As already proposed for DNA translocation through α-hemolysin[26], this extra resistance is likely originated by the enantiomeric properties of the desoxi-D-ribose, which produces more steric hindrance for DNA bases as they move down the channel encountering large side chains (e.g., R220, R282) that render the lumen narrower (Fig. 2b). Therefore, we conclude that, at the low applied voltages (approx. 200 mV) and high ion concentration (1–3 M KCl), the extra negative charge at the 5' end is not enough to counterbalance the additional energy barrier needed to translocate ssDNA by the 5' end.

**Aerolysin nanopore presents two main sensing spots**. Next, to precisely map the sensing spots of aerolysin for single-molecule detection, we designed a series of oligonucleotides containing an abasic site (marked by an "X", which means that the nucleobase is substituted by a hydrogen atom[32], Fig. 3a) at different positions of $dA_{14}$. It should be noted that we used free ssDNA, which is

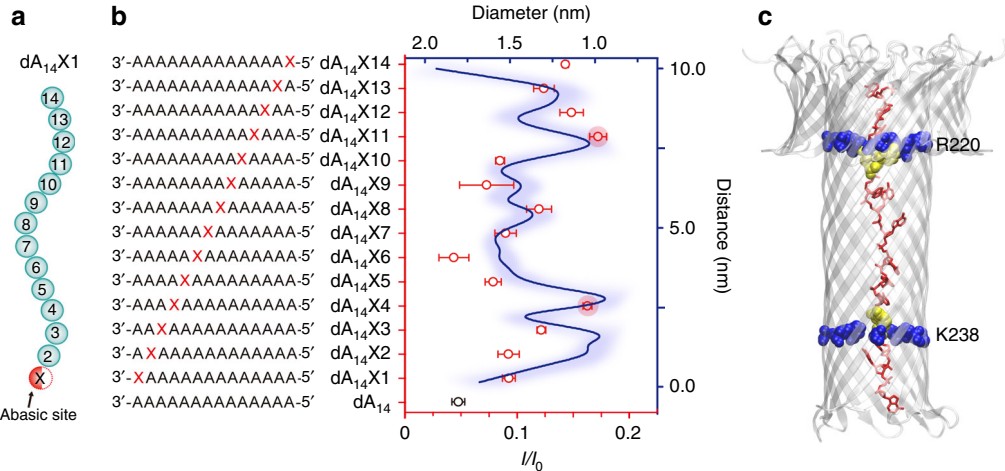

**Fig. 3** Mapping the sensing spots of aerolysin by abasic site scanning of $dA_{14}$. **a** Schematic representation of $dA_{14}X1$ oligonucleotide; the red sphere of X represented the single-nucleobase deletion. **b** Comparison between the percentage of increase in residual current for each abasic oligonucleotide (red circle with error bar) and the pore diameter (blue line) calculated by the PoreWalker server. Data were obtained in 1.0 M KCl, 10 mM Tris, and 1.0 mM EDTA, pH=8.0, 24 ± 2 °C at the bias potential of +140 mV. The errors of the data were based on at least three separate measurements. **c** Selected MD snapshot of $dA_{14}$ fully inserted in the pore, showing the two main constriction points (R220 and K238, in blue van der Waals representation) and the DNA in red, with bases at positions 11 and 4 in yellow van der Waal representation

different from previously reported methods that were using biotin–streptavidin complex or hairpin structure to immobilize ssDNA[32,33]. In comparison to them, our free ssDNA could precisely reflect the dynamic properties of the system. The sequencing of oligonucleotides is depicted in Fig. 3b and are named as $dA_{14}Xn$ (with $n$ from 1 to 14), in which the position was numbered relative to the 3′ end of the oligonucleotide. The residual current data for each oligonucleotide can be found in Supplementary Figs. 4–18. All $I/I_0$ histograms are fitted to a Gaussian function and all abasic oligonucleotides presented higher residual current values than $dA_{14}$ (Fig. 3b, red circle with the error bars), which is consistent with previous studies of α-hemolysin[32], as the absence of a nucleobase induced a smaller blockade of ion current. Notably, a maximum value of $I/I_0$ occurred when X was located at positions 4 or 11, suggesting the existence at those positions of two sensing spots across the aerolysin pore.

Calculating the diameter of the aerolysin pore using PoreWalker[34], we observed a direct correlation between the diameter of the lumen and the increase in ion flux for abasic ssDNA translocation. The two main constriction points, where the inner region of the β-barrel becomes <1 nm in diameter, and which are located around R220 and R238, coincide with the putative position of the abasic site in $dA_{14}X11$ and $dA_{14}X4$, respectively (Fig. 3b). Analyzing the MD simulation of $dA_{14}$ fully inserted in the pore, we observed that the 11th base is in fact located at the heptameric ring formed by R220, and that the 4th base of $dA_{14}$ is interacting with the ring formed by K238 (Fig. 3c). Therefore, the two main aerolysin sensing spots would correspond to amino acids R220 and K238. Interestingly, during MD, transient π-stacking interactions were observed around both constriction points, between the 11th and the 12th bases, or the 3th and 4th. Similar interactions have been previously described as related to ionic current blockade in the MspA pore[35].

**Discrimination of single different nucleobases.** Once these two sensing spots were identified, we aimed to study their ability for discriminating different nucleobases. To do so, we replaced the adenine nucleobase in positions 4 and 11, one at a time, to cytosine, thymine, and guanine (called hereafter $dA_{14}X4$-C, $dA_{14}X4$-T, $dA_{14}X4$-G, $dA_{14}X11$-C, $dA_{14}X11$-T, and $dA_{14}X11$-G,

respectively). Unlike in α-hemolysin, the lack of a vestibule structure at the pore entry and the narrower pore diameter makes it in principle much harder for aerolysin to capture long ssDNA (>10 bases)[36]. Therefore, we used a higher salt concentration of 3.0 M KCl to increase the aerolysin capture rate, in order to obtain enough data for statistical analysis. We added the oligonucleotides one by one with the same concentration (3.0 μM) in the *cis* chamber and measured the residual currents. For the sensing spot at position 4, we observed an overlap in residual current values between both $dA_{14}X4$-G and $dA_{14}X4$-A, and also between $dA_{14}X4$-T and $dA_{14}X4$-C (Supplementary Fig. 19). On the contrary, for position 11, all the residual current histograms fitted to a Gaussian distribution and showed a perfect current separation (Fig. 4a). The four nucleobases presented a residual current ranked as G < A < T < C, which is consistent with the results obtained for single-nucleobase discriminations by the $MoS_2$ nanopores[37], and is well correlated to nucleobase size.

Moreover, taking advantages of the optimal current separation between $dA_{14}$ and $dA_{14}X11$-C/G, we further exploited the pore ability for detection of modified nucleobases in a mixture sample containing methylated cytosine ($^mC$) and oxidized guanine ($^oG$). Sensing position 11, defined by the constriction at R220, has indeed the ability to identify all four kinds of nucleobase (A, T, C, G, Fig. 4a) and two modified nucleobases ($^mC$ and $^oG$, Fig. 4b). Finally, we studied the ability of aerolysin to identify different single nucleobases in a heteropolymeric ssDNA strand, using four kinds of oligonucleotides that only differ at position of 11 (Fig. 4c). In contrast to the results of $dA_{14}$ homopolymeric analysis, the order of residual current of A and G changed for heteropolymeric DNA, which demonstrates that the flanking nucleobase significantly affect the streaming of blockade current levels.

The ionic current was calculated using a well-established method[35,38] for an aerolysin pore system without ssDNA and for the $dA_{14}$ system, giving results of 118.6 ± 1.4 and 0.7 ± 0.3 pA, respectively (mean values (± s.d.) in three separate simulations). Our approach, however, is known to overestimate the absolute current mainly due to the overestimation by the CHARMM force field of the bulk electrolyte conductivity by ~10–40%[35,38,39,40]. Given the approximate nature of our approach, the calculated

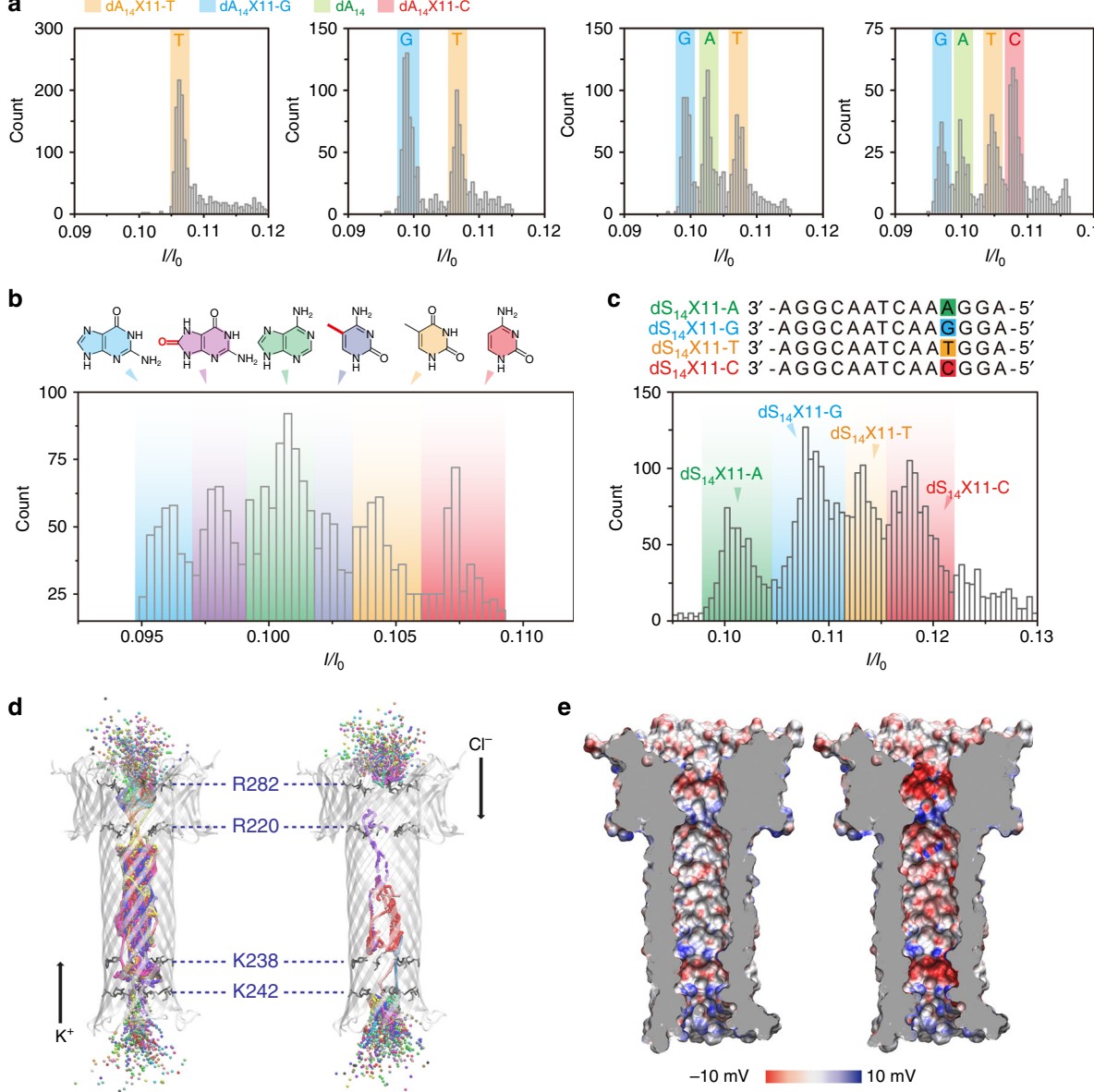

**Fig. 4** Discrimination of all four DNA bases by $dA_{14}X11$ sites of the aerolysin nanopore. **a** The $I/I_0$ histograms of $dA_{14}X11$-T; $dA_{14}X11$-T and $dA_{14}X11$-G; $dA_{14}X11$-T, $dA_{14}X11$-G, and $dA_{14}$; $dA_{14}X11$-T, $dA_{14}X11$-G, $dA_{14}$, and $dA_{14}X11$-C, from left to right. The number of blockades in each histogram is at least 2000. **b** The $I/I_0$ histograms for aerolysin pore interrogated with six kinds of oligonucleotides that only differ at the position of 11 (four kinds of normal base: A, T, C, G and two kinds of damage nucleobases: $^mC$ and $^oG$). The number of blockades is 4500. **c** Histograms of $I/I_0$ caused by the four kinds of nucleobase in a heteropolymeric DNA strand. The sequence information of four ssDNA is illustrated at the top of the figure. The data were obtained for 3.0 M KCl, 10 mM Tris, and 1.0 mM EDTA, pH=8.0, 17 ± 2 °C at the bias potential of +100 mV. The number of blockades in each histogram is at least 5000. **d** Potassium (left) and chloride (right) ion occupancy at each 0.1 ns of the MD simulation. The DNA ($dA_{14}$) is not shown for sake of clarity. Ions are represented as spheres colored according to residue ID. Amino acids R282, R220, K238, and K242 are shown as black sticks. **e** Electrostatic surface of aerolysin pore lumen as calculated with the APBS-PDB2PQR server[59,60], without (left) or with (right) $dA_{14}$ inserted in the pore. The DNA is not shown for the sake of clarity

current is thus reasonably well in line with the experimental values (125.0 and 5.0 pA) (Supplementary Fig. 20). Looking at the distribution of ions along the pore during MD simulations with $dA_{14}$ (Fig. 4d), we observed large ion occupancy in between the two constriction points, while ions were more confined around R220/R282 and K238/K242, likely contributing to determine the specific ion current features of the aerolysin pore. Furthermore, while DNA translocates through the pore, the electrostatic potential in the lumen becomes more negative, particularly around the two constriction points (Fig. 4e), affecting the ion

current; namely negative and larger chloride ions are more easily blocked when compared to potassium ions (Fig. 4d).

## Discussion

Residual current and duration time of translocation are the two most significant properties that can be used to discriminate different DNA sequences translocating across nanopores. Understanding these two characteristics at a molecular level is of critical importance for the design of new and more sensitive nanopores. Herein, we present an analysis of ssDNA translocation across

aerolysin biological pores, which is based on both experimental and computational analysis, and allows us to unveil novel properties of aerolysin-oligonucleotide interplay.

Previous works about DNA nanopore sequencing using PFTs show the importance of the pore diameter for resolution. For example, the MspA nanopore (~1.2 nm diameter) is capable of higher current resolution than α-hemolysin (1.4 nm diameter) due to its smaller diameter funnel-like aperture. Herein, we show that the diameter of aerolysin channel is even smaller, with a value under 1 nm at its narrowest constrictions, which can be related to its observed high sensitivity for analytes, including oligonucleotides[18] and poly(ethylene glycol) molecules[17]. Single-channel measurements in different lengths of $dA_n$ demonstrated that $dA_{14}$ performed a full blockade of aerolysin nanopore, and computational studies suggested that the optimal steric and electrostatic match of $dA_{14}$ length with the pore barrel extension is likely at the origin of the observed increased duration time of $dA_{14}$ translocation. MD simulations showed relevant salt bridge interactions between both the 5′ and 3′ DNA phosphates and R282 and K244 residues at the pore entry and exit, respectively, which are likely responsible for capturing ssDNA for effective translocation. Comparing the pore lumen of aerolysin with the other commonly used PFT-based nanopore, α-hemolysin, and MspA, we observe a greater abundance of charged residues in aerolysin, mainly at the pore exit, which can favor translocation (Fig. 4e and Supplementary Fig. 21). As mutational experiments of charged amino acids in other PFTs have been successfully performed to precisely control the translocation capabilities[21,41], aerolysin presents a significant potential to further improve its native sensing capabilities.

In particular, based on the long translocation time of $dA_{14}$, longer than in any other pores, we could use this oligonucleotide construct as a quasi-immobilized probe, as similarly achieved by DNA-streptavidin based protocols used to study other faster-translocating nanopores[33,42]. Thus, experiments of the abasic site at different positions in $dA_{14}$ were able to precisely map the sensing spots of aerolysin at single-nucleotide resolution. The complementary computational analysis allowed us to identify, as also expected from the initial structural analysis of the pore, R220 and K238 as the key sensing residues, providing valuable information to understand the high sensitivity of aerolysin for oligonucleotide detection. Our results show a good relationship between ion flux decrease and the pore diameter, suggesting that the main determinant for the residual current is produced by pore steric hindrance, as already proposed for other nanopores[35,43,44]. The narrowest constriction defined by R220 is most sensitive for the discrimination of all four types of nucleobases, as well as cytosine methylation and oxidation of guanine in a free ssDNA. In comparison to α-hemolysin and MspA, the sensing spot produced by a heptameric ring of large arginine residues is significantly narrower (less than 0.9 nm), allowing aerolysin to detect dinucleotides, which is not feasible for the two other pores mentioned above.

To achieve the high sensitivity, nanopore technology has been searching for nanopores with a translocation height as small as one nucleotide in order to decrease the sequencing noise. In addition, the first studied α-hemolysin pore, with a translocation height of 5.0 nm, has been lately replaced by MspA, with a height of less than 0.6 nm[11]. Notably, the aerolysin pore is constituted by a barrel longer than other PFTs, reaching nearly 10 nm. Our study reveals that the actual sensing height of aerolysin is indeed one amino acid at its narrowest constriction point (R220), which is associated with a longer pore lumen that could be the reason for the long translocating time, which is also necessary for an improved resolution. Moreover, the longer pore lumen, together with the existence of several charged residues mainly at the pore

exit, opens up the opportunity to modulate aerolysin sensitivity by site-directed mutagenesis, creating nanopores with controlled specific functions. Interestingly, in MD simulations we observed the remarkable flexibility of the aerolysin barrel, which could pass from circular to elliptical conformations. This native flexibility may help to better translocate molecules along the pore and should be considered to understand and design the aerolysin pore variants.

Altogether, the data presented here allow setting the bases for performing sequence modifications in aerolysin, which will hopefully yield nanopores with improved sensitivity and resolution. While this manuscript was revised, a work showing the potential of aerolysin for peptide sequencing was published, which further strengthens the potential of this pore for future applications[45]. Moreover, nanopore techniques could be further developed as a single-molecule method to investigate the function or structure of membrane pore in real biological environments.

## Methods

**Materials**. Details about the materials used in this study can be found in the Supplementary Methods.

**Single-channel recording experiments**. 1,2-Diphytanoyl-*sn*-glycero-3-phos-phocholine powder (Avanti Polar Lipids, Alabaster, AL, USA) was dissolved in decane (Sigma-Aldrich, St. Louis, MO, USA) for a final concentration of 2.0 mg per 100 μl. Proaerolysin protein was incubated with Trypsin-EDTA (Sigma-Aldrich, St. Louis, MO, USA) for 10 h at room temperature and then stored at −20 °C. Lipid bilayer membranes were formed across an orifice with the dimeter of 50 μm in a Delrin bilayer cup (Warner Instruments, Hamden, CT, USA), which separated the recording apparatus into two chambers, *cis* and *trans*. After the addition of aero-lysin monomer protein to the *cis* chamber, it could self-assemble to form a transmembrane heptameric protein pore. A single aerolysin nanopore generated a constant current of nearly 50.0 pA at baseline under the applied voltage of +100 mV in at a temperature of 24 ± 2 °C in 1.0 M KCl solution buffered with 10 mM Tris and 1.0 mM EDTA, titrated to pH 8.0. DNA oligomers were added to the *cis* side of the pore. Two matched Ag/AgCl electrodes were used to record the ionic currents. Then, the current traces were amplified and measured with a patch clamp amplifier (Axon 200B) equipped with a Digidata 1440A A/D converter (Molecular Devices, Sunnyvale, CA, USA). The signals were filtered at 5 kHz and acquired with Clampex 10.4 software (Molecular Devices, Sunnyvale, CA, USA) at a sampling rate of 100 kHz. The data were analyzed using MOSAIC[46] and OriginLab 8.0 (OriginLab Corporation, Northampton, MA, USA) software. The DNA sequences used in this study are listed in Supplementary Table 2.

**Molecular modeling and simulations**. The aerolysin pore was described using the equilibrated conformation defined by previous MD simulations of the aerolysin pore[47]. This original model was reduced by removing the membrane binding domains (i.e., residues 24–195, 301–408, and 425–447), and the last strand (residues 409–424) was linked to the previous one by creating a loop between amino acids A300 and Q409 using the software Modeler v9.13[48]. The simulations were thus performed on the membrane spanning β-barrel only, which remained stable as the full pore unit during all the MD simulations presented here. This same approximation has been already used for translocation studies in other PFTs[28,49]. A ~11 × 11 nm² membrane bilayer was modeled by 1-palmitoyl-2-oleoylphosphatidylcholine (POPC) lipids using the charmm-gui server, following the membrane positioning as suggested by the PPM server[50]. A ssDNA was derived from a model of double-stranded DNA of 10, 14 or 20 adenosines, created with the 3D-DART web server[51]. A phosphate group at 5′ end position, with two negative charges, as predicted by the chemicalize server (https://chemicalize.com/), was parameterized with the CGenFF tool of the charmm-gui web server[52]. Afterwards, the ssDNA molecule was manually placed on the top of the barrel (extracellular side), pointing either its 5′ or 3′ terminus to the pore entry, depending on the studied direction of entry. This system was then solvated in a 1 M KCl water box with initial dimensions ~11 × 11 × 21.7 nm³.

All MD simulations were run using the GROMACS software version 4.8[53], with the charmm36 force field[54], the SHAKE algorithm on all the bonds between hydrogen and heavy atoms, and Particle-Mesh Ewald, treating the electrostatic interactions in periodic boundary conditions. The system was first minimized using the steepest descent algorithm, and afterwards equilibrated using a similar protocol of that suggested by the charm-gui server[55], but in 2 blocks of 6 steps each, totalizing 975 ps. In the first block of equilibration, the lipid restraints were gradually reduced to zero, while those on protein and DNA were maintained, and were only gradually and completely removed in the second block. Afterwards it was simulated for 10 ns without restraints. A more detailed description of this

equilibration protocol can be found as Supplementary Table 3. The approximate dimensions of the box at the end of the equilibration were ~10.6 × 10.6 × 21.3 nm³.

For all further simulations (SMD and MD of selected snapshots), we chose an integration step of 2 fs. A temperature of 22 °C was controlled with the Nose-Hoover thermostat and the Parrinello-Rahman method was used for semi-isotropic pressure coupling. SMD was used to introduce the DNA in the aerolysin pore, using an umbrella biasing potential, based on direction-periodic geometry and using as reaction coordinate the z axis, with the aerolysin pore aligned to it. A harmonic biasing force (spring constant of 100 kJ mol⁻¹ nm⁻²) was applied to the spring connecting the center of mass of either the 3′ or the 5′ nucleotide, depending on the direction of entry studied, and the center of mass of the α-carbons of the seven K244 residues, located at the pore exit, at a constant velocity of 0.004 nm ps⁻¹. A biased voltage of 250 mV was applied. The stretching events during SMD, and its relaxation during MD, have been measured based on the change of the inter-strand distance, as already performed by others[56] (Supplementary Fig. 22). A full relaxation of the DNA stretching produced by SMD is observed already in the first ns of unbiased MD simulations; thus it reaches stabilization of the initial configuration with a length of 73 Å, lower than the estimated contour length for dA₁₄, given that inter-phosphate distances of ssDNA can range between 5.9 and 7 Å[57,58].

For the study of dA₁₄ fully inserted on the aerolysin pore, three snapshots were selected from three independent replicas of SMD, where the DNA is located with its 3′-end at the pore exit and its 5′-end at the pore entry. For the study of shorter (dA₁₀) and longer (dA₂₀) oligonucleotides, two SMD snapshots were selected for each DNA length, one with the 3′-end located at the pore exit, and another with the 5′-end located at the pore entry. Those selected snapshots were then equilibrated for 150 ps, using the former pulling protocol but with constant velocity equal to zero, to keep the pulling distance unchanged, and afterwards submitted to 100 ns of unrestrained MD simulation, applying a biased voltage of 250 mV. All replica of dA₁₄ showed the same positioning and interactions, and therefore only one of them was further simulated until a total time of 250 ns. The protein remained very stable during simulations at this electric field without the need of applying position restraints. This is likely due to the prion-like stability conferred to the pore by the concentric double β-barrel conformation at the extracellular end (see Supplementary Fig. 23 for RMSD plots of the equilibration and MD simulation at applied voltage). In all simulations with applied electric field, the dimension of the box along the z axis was kept fixed. An image showing the location of the protein with respect to the lipid bilayer membrane and the extent of the solvent volume at the end of the equilibration has been added in Supplementary Fig. 23.

A snapshot from the MD of dA₁₄ fully inserted in the pore was used for electrostatic calculations using the APBS-PDB2PQR web server[59,60], with and without ssDNA at 1 M salt concentration. The ion current was calculated following a method already described by others[38] using the following formula, where $z_i$ and $q_i$ are the z coordinate and the charge of atom $i$, respectively:

$$I(t) = \frac{1}{\Delta t L_Z} \sum_{i=1}^{N} q_i [z_i(t + \Delta t) - z_i(t)] \qquad (1)$$

This calculation was performed similarly for two systems, one with dA₁₄ fully inserted in the pore, and the other with no DNA. The free-DNA system was simulated following the same procedure described above for the dA₁₄ system, applying an electric field corresponding to a voltage of 250 mV. The systems were simulated for 250 ns, but only the last 200 ns were used for the calculations (as to allow DNA to adjust its position from that obtained from SMD). Our approach is known to overestimate the absolute current by 10–40%, mainly due to the overestimation of KCl bulk conductivity by the CHARMM force field[35,38,39,40]. Results are provided in Supplementary Fig. 20.

**Data availability**. All relevant data included in the paper and its supplementary information are available on request.

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

## Acknowledgements

This research was supported by the National Natural Science Foundation of China (21421004, 21327807), the Program of Introducing Talents of Discipline to Universities (B16017), Innovation Program of Shanghai Municipal Education Commission (2017-01-07-00-02-E00023), the Fundamental Research Funds for the Central Universities (222201718001, 222201717003), the Swiss National Science Foundation (to M.D.P.), the European Union's Horizon 2020 Research and innovation program under the Marie Skłodowska-Curie Grant Agreement No. 665667 (to C.C). N.C. is supported by a fellowship of the Conselho Nacional de Desenvolvimento Científico e Tecnológico (CNPq) of Brazil. We thank WenPing Lyu for discussion of current calculations, and Matteo Degiacomi for discussion on aerolysin molecular simulations.

## Author contributions

C.C., H.T., and Y.T.L. conceived the idea; C.C., M.-Y.L., and Y.-Q.W. designed and performed the nanopore experiments, collected and analysis data; C.C. and N.C. performed the molecular dynamics simulations; C.C. and M.-Y.L. performed image processing; C.C., N.C., M.D.P., and Y.-T.L. interpreted the data; C.C., N.C., M.D.P., and M.-Y.L. wrote the paper; Y.-T.L. supervised the project; M.D.P. supervised molecular modeling and simulations for the project and all authors revised the paper.

## Additional information

**Competing interests:** The authors declare no competing interests.

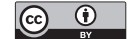

