## [Peer Review File · Nature Communications]

Reviewers' comments:

Reviewer #1 (Remarks to the Author):

This high-quality manuscript describes experimental and computational studies of the biological nanopore aerolysin for the purpose of DNA sequence sensing. First, the authors investigate how the ionic current blockades depend on the length of the poly(dA) strand, finding that increasing the length of the strand beyond 14 nucleotides does not change the amplitude of the blockade ionic current. Next, the authors find that a DNA strand is more likely to insert with its 3' end first than with the 5' end, which is consistent with the previous observations made using the alpha-hemolysin nanopore. By introducing an abasic residue at different locations along a poly(dA)₁₄ strand, the authors identify two regions of the nanopore that most likely affect the ionic current. The authors directly verify their assessments by introducing single nucleotide substitutions at the positions identified using the abasic nucleotide measurements. Amazingly, the authors find the residual current through aerolysin to very sensitively depend on the type of the DNA nucleotide located at position 11 of the strand, which allows them to not only differentiate all 4 DNA nucleotides but also two modified varieties. Distinct ionic current levels were also observed in experiments performed using heterogeneous sequence DNA containing a single nucleotide substitution, although the currents were also found to depend on the sequence of the flanking fragments. The authors rationalize some of their findings through all-atom molecular dynamics (MD) simulations of the experimental systems.

Overall, this is a very interesting manuscript reporting an important development in the nanopore sensing field. Although several other nanopore systems were used to determine the nucleotide makeup of a DNA strand by measuring the nanopore ionic current, achieving single nucleotide sensitivity required arresting the electrophoretic transport of DNA using biological enzymes. The present manuscript shows that single nucleotide discrimination can be obtained in the absence of enzymes controlling DNA transport, which is an important achievement. The manuscript is clearly written and can be understood by a non-specialist.

There are three general areas where the manuscript can be improved:

1. It is not clear whether the single nucleotide discrimination is possible for DNA fragments that are not exactly 14 nucleotide long. The fact that the authors were able to observe clear ionic current signatures from individual nucleotides suggests that the time the DNA strand spends entering and leaving the nanopore is negligible compared to the residence of a full 14-nucleotide fragment. So what would happen to a DNA that is, say 16 nucleotides long?
2. The protocols used to carry out MD simulations and interpretation of some of the MD results is questionable. Please see more detailed comments below.
3. Some critical references to prior studies are either missing or used incorrectly.

Specific comments:

Lines 107-109. State upfront the duration of the MD simulation and hint how the DNA was introduced into the nanopore.

Lines 118-120 and lines 280-291: The authors say that MD result (Figure 1f) suggest "optimal placement" of a poly(dA)₁₄ fragment. This cannot be a correct statement as the authors have simulated only the 14-nucleotide fragment. To conclude about optimality of such a placement that authors should also simulate a shorter and longer fragment and arrive at the conclusion through comparison of the three cases.

Figure 1b. What is the ionic current signature associated with DNA exit from the nanopore? The current traces presented leave an impression that the pore becomes permanently blocked. Please

add exit signatures to the plots.

Lines 134-182 and Figure 2. It appears that the authors missed altogether the previous study by Mathe and co-workers that investigated orientation-dependent transport of DNA hairpin through alpha-hemolysin [PNAS 102:12377]. The authors cite this study but out of its context. For example, Figure 2b of that manuscript shows that entry probability is higher for 3' insertions whereas Figure 6 details the microscopic mechanism derived from MD simulation. Does the mechanism described in Figure 2b of the present manuscript differ substantially from that described 12 years ago?

Caption to Figure 3b and Figure 4a-c. Please specify the number of events used to assemble the histograms.

Line 227: Why increasing the ion concentration to 3M increases the DNA capture rate?

Line 228: The concentration of DNA in the sample (3 micromolar) appears to be a lot higher than used in previous experiments with biological pores. Please comment on that.

Figure 4a-c: Please label the histograms using the names of the substituted nucleotides. Do not rely only on the color code as some people have problems with seeing different colors and/or use black and white printers.

Figure 4e: make the electrostatic axis scale in the units of mV.

Lines 249-252. The authors use the ionic current simulations to identify the region of the pore that modulates the current most. In doing so, they conclude that "ion flux was clearly limited around select residues". Well, the ionic current through any cross-section of the channel must be constant to satisfy the current continuity requirement. The only situation in which the current through different sections of the channel can be different is if the simulation has not yet reached a steady state regime. In that case, the simulation outcome will depend on initial conditions.

Line 311: The claim of "de facto" better sensitivity must be supported by signal to noise analysis.

The MD methods sections and the results has a number of problems, some of them are listed below:

Overall, the authors do not provide enough details for someone to reproduce the simulation and do not characterize the outcome of the simulation with the rigor common to the field.

Line 355: A 5' phosphate can be added to DNA using a standard CHARMM patch, there is no need to do parameterization.

The authors appear to miss the following two critical MD studies that examined the effect of arginine mutants on DNA transport through MspA [ACS Nano, 6:6960] and determined the molecular origin of the sequence specific ionic current blockades [ACS Nano 10: 4644]. The authors are asked to discuss the result of their work in the context of these studies.

Line 369. The equilibration must be performed in the NPT, not NVT ensemble. NPT ensemble is required for the system to acquire equilibrium dimensions and to reach the target pressure of 1 bar.

Line 370: Specify the spring constant of the restraints and duration of each equilibration step.

Line 374: Using SMD to introduce a DNA strand into biological nanopore is not the best idea. The application of the force to the leading nucleotide will stretch the DNA strand, which could lead to

artificially elongated DNA conformation in the nanopore. A much more superior approach is in using a Grid-Steered MD, see [JCP 127:125101] for details. In any event the amount of stretching must be detailed in SI.

The authors carried out several simulations of the system at different voltages but they never report the actual simulated current. What was the open pore and blockade currents obtained using the MD methods? Please add integrated current plots to the SI.

Was the protein structure restrained during the E-field simulation?

Reviewer #2 (Remarks to the Author):

The authors in this article report the use of a group of short polydeoxyadenines (dAn, n=10-20) and single-base substitution variants to pin-point the sensing spots in an aerolysin nanopore. The purpose of this work is to demonstrate the potential for nanopore sequencing. In the key experiment, the DNA dA14 was selected to perform the abase screening. The abase constructed can slightly change the amplitude of the translocation current block depending on the abase location in the DNA. Assuming in a conformation in which dA14 is fully trapped in the pore during translocation, the abase location-dependent current amplitude was used to map the sensing spot, which is considered to correspond to the abase site with the highest current amplitude. Interestingly, different base substitutions at that site can influence the block current differently, allowing discriminating dA14 variants with A, T, G, C and methylated C substitutions.

Indeed, this work demonstrates the potential of aerolysin for precise detection of short oligonucleotide. However, due to the intrinsic structure of aerolysin, its impact on DNA sequencing is not high. As neighboring bases in a DNA are separated smaller than 1 nm, the nanopore must contain a very thin sensing motif to accurately call the identity of each base in sequencing. It has been commonly recognized that a nanopore possessing an outstanding thin and narrow constriction (e.g. funnel or cone shape) can be applied for sequencing, because in these pores only the constrictive site controls the current flow. Currently MspA and CsgG that meet this structure requirement have been successfully developed for sequencing. In contrast, aerolysin forms a long and narrow drum barrel. In this structure, as all parts of the pore including the "sensing spots" contribute to the current flow, it is difficult to read each base as a long DNA passes through the pore. This structural disadvantage lowers the enthusiasm for aerolysin sequencing. The authors published an excellent work on discriminating polydeoxyadenines of different lengths in aerolysin (Nat Nanotech 2016), but these DNAs are much shorter than the pore length.

In addition to the significance issue, several mechanistic problems are listed below

In Figure 3, dA14 was used for single-abase screening. Indeed, the current amplitude for dA14 translocation can be changed with the abase position. However, it is not convincing to utilize the current change to map the sensing spot. Each current block observed represents an entire procedure for DNA dynamic translocation, which involves every base that enters, slides and leaves the pore. This process must include many conformations for DNA in different positions in the pore (including partial entering the pore). That is to say, Figure 3b and c shows only one conformation that has a dA14 fully trapped in the pore during translocation. The experiments in the paper lack convincing evidence to justify how this current is generated, and whether the current amplitude for translocation is generated by this illustrated conformation or other conformations. Therefore it is impossible to map the sensing spots only based on the abase DNA current amplitude.

In Figure 2, dA14 was used to study the translocation direction. By attaching a streptavidin to block one end, dA14 can only enter the pore from the other end. Both translocation directions can be identified by two current amplitudes in the histogram (Figure 2a). However, the current

distributions for all other DNAs (dA10 to dA20) in Figure 1c and d only show a single current component. This controversy causes confusion, which one is correct? Can all DNAs translocate in two directions? Furthermore, can dA14 abase variants translocate in both directions? Please verify experimentally. If they do, there should be two set of I/I₀ data for each abase variant that should be shown in Figure 3b to map the pore amino acids. Also the component assignments in histograms in Figure 4 would be more complicated for base discrimination.

dA14 is a special DNA. According to the model shown in the paper, it is about the same length as the aerolysin pore, 10 nm. Also its block duration is the longest as shown in Figure 2e. It was interpreted that "the interaction between dA14 and the aerolysin nanopore is stronger, likely due to the optimal spatial filling of dA14 within the ~10 nm long pore channel," and "This optimal steric and electrostatic match 119 of dA14 within the lumen is likely at the origin of the observed increased duration of 120 translocation time." While this explanation is acceptable, why do longer DNAs (dA15 ~ dA20) that contain the same dA14 domain and experience the same interaction in the pore have much shorter translocation duration? Can longer dA15~dA20 get the similar abase screening result and single base sensitivity as dA14. Can the conclusion drawn from dA14 such as single base sensitivity and sensing spots be generalized to long DNAs? This has not been evidenced yet and limited the impact.

In addition, there are minor issues

Figure 1b needs to show the entire block events, rather than the initial parts, for all polydA DNAs, such that the current profile of the blocks can be analyzed.

Figure 4a, b and c should use I/I₀ to represent the current blocks in x-axis, to keep consistence with and compare with current amplitude in other figures.

Response to the Reviewer #1

Thank you very much for your constructive suggestions and your detailed comments, which have been very helpful towards improving our manuscript. We have put significant effort to revise and improve our manuscript on these three aspects: (i) we have complemented the study with other DNA fragments (dA_{10} and dA_{20}) investigated both experimentally and computationally; (ii) added more detailed information regarding MD simulations; (iii) included important missing references to support our discussion.

The revised parts are marked in red-colour font in the main text. A detailed, point-by-point response is reported below:

Q1. Lines 107-109. State upfront the duration of the MD simulation and hint how the DNA was introduced into the nanopore.

A1: We have now included this information in the manuscript (Page 3):

“To prove this hypothesis and better characterize the main interactions between DNA and aerolysin, which may explain the longer duration time observed for dA_{14} , we performed 250 ns of molecular dynamics (MD) simulations of dA_{14} fully inserted in the pore lumen. The initial conformation was obtained by using steered MD, as explained in the methods section.”

Q2. Lines 118-120 and lines 280-291: The authors say that MD result (Figure 1f) suggest “optimal placement” of a poly(dA)₁₄ fragment. This cannot be a correct statement as the authors have simulated only the 14-nucleotide fragment. To conclude about optimality of such a placement that authors should also simulate a shorter and longer fragment and arrive at the conclusion through comparison of the three cases.

A2. We agree with the reviewer and we have now performed simulations of dA_{20} and dA_{10} , following the same procedure used for dA_{14} . For the longer dA_{20} , the results showed that this oligonucleotide remains in fact longer than the pore length, resulting, depending on starting conditions, in the accumulation of the remaining bases at the pore entry or exit (see Figure S2). For dA_{10} , we observed as expected that it does not completely fill the pore, and the DNA tends to accumulate at the pore exit. The sampling of these MD simulations does not allow us to draw any definitive quantitative description of the translocation process (as here they are mostly used as an exploratory means to interpret and guide experiments), however, these additional conformations are suggesting that dA_{14} is trapped for longer time in the pore because its optimal match with the length of the channel, which might block for longer time ions translocation. This is not the case for shorter oligonucleotides, that have already the tendency to exit the pore, and for longer, which having a portion always exposed to the bulk can facilitate ions translocation and accelerate the exit of the ssDNA from the pore. A discussion regarding this aspect has been added to the revised manuscript (see Pages 3-4 and Supplementary Figure S2):

*“This observation agrees with previous MD studies using the MspA pore, which revealed that the inclusion of extra positive side chains in the lumen resulted in an increase in translocation time²⁶. The optimal steric and electrostatic match of dA_{14} within the lumen is likely at the origin of the observed increased duration of translocation time. By comparison, shorter (dA_{10}) and longer (dA_{20}) oligonucleotides did not show the same perfect match during MD simulations (**Supplementary Figure S2**). For the longer dA_{20} , the results showed that this oligonucleotide remains in fact more extended than the pore length, resulting, depending on starting conditions, in the accumulation of the remaining bases at the pore entry or exit. On the other hand, the shorter dA_{10} does not completely fill the pore, and the DNA tends to accumulate at the pore exit. Contrary to the optimal match of dA_{14} , shorter or longer*

oligonucleotides may accelerate their exit from the pore by facilitating a faster restoration of ion current thanks to conformations that allow for a partially filled pore lumen (e.g., dA₁₀) or solvated bases in the bulk (e.g., dA₂₀)."

Q3. Figure 1b. What is the ionic current signature associated with DNA exit from the nanopore? The current traces presented leave an impression that the pore becomes permanently blocked. Please add exit signatures to the plots.

A3. We have now added the exit signature plots of dA_n typical blockade current in the new Figure 1b.

Q4. Lines 134-182 and Figure 2. It appears that the authors missed altogether the previous study by Mathe and co-workers that investigated orientation-dependent transport of DNA hairpin through alpha-hemolysin [PNAS 102:12377]. The authors cite this study but out of its context. For example, Figure 2b of that manuscript shows that entry probability is higher for 3' insertions whereas Figure 6 details the microscopic mechanism derived from MD simulation. Does the mechanism described in Figure 2b of the present manuscript differ substantially from that described 12 years ago?

A4. We apologize but we actually missed it and we have now added it and properly discussed in the revised manuscript at Pages 5-6. The mechanism we found for aerolysin is similar indeed to what found previously for α -hemolysin.

"As already proposed for DNA translocation across α -hemolysin²⁴, this extra resistance is likely originated by the enantiomeric properties of the desoxy-D-ribose, which produces more steric hindrance for DNA bases as they move down the channel encountering large side chains..."

Q5. Caption to Figure 3b and Figure 4a-c. Please specify the number of events used to assemble the histograms.

A5. We have now added the number of events used to assemble the histograms in the caption of Figure 3b and Figures 4a-c.

Q6. Line 227: Why increasing the ion concentration to 3M increases the DNA capture rate?

A6. The capture rate (f) is calculated by $1/t_{on}$, while t_{on} is the interval time between two adjacent blockade events. DNA has to overcome an energy barrier as it is captured and transported through a biological nanopore, according to previous studies in α -hemolysin this barrier will be decreased with a high salt concentration¹. The same appears to be valid also for aerolysin. As we calculated the capture rate of dA₁₄ at 1M, 2M, and 3M KCl solution, with the same DNA concentration (see Figure R1, review only), the results indicated that the capture rate of dA₁₄ increases 15 times at 3M KCl (12.31 s^{-1}) when compared to 1M KCl (0.8 s^{-1}).

Figure R1. Raw current trace of dA₁₄ transported through aerolysin nanopore at 1M KCl (a), 2M KCl (b), and 3M KCl (c). The concentrations of dA₁₄ are the same in all experiments (5 μ M in the *cis* chamber).

Q7. Line 228: The concentration of DNA in the sample (3 micromolar) appears to be a lot higher than used in previous experiments with biological pores. Please comment on that.

A7. During our study, we noticed that it is difficult for aerolysin to capture longer oligonucleotides (>10 bases). Recently, a work also reported that it is a challenge to capture long DNA/RNA efficiently². We have added a comment in the main text to comment this aspect: *“Unlike in α -hemolysin, the lack of a vestibule structure at the pore mouth and the narrower pore diameter makes it in principle more difficult for aerolysin to capture long ssDNA (>10 bases)³⁴. Therefore, we used a higher salt concentration of 3.0 M KCl to increase the aerolysin capture rate, in order to obtain enough data for statistical analysis.”* (Page 8)

Q8. Figure 4a-c: Please label the histograms using the names of the substituted nucleotides. Do not rely only on the color code as some people have problems with seeing different colors and/or use black and white printers.

Figure 4e: make the electrostatic axis scale in the units of mV.

A8. We have now labelled the different substituted nucleotides and changed the electrostatic axis scale in the units of mV (see new Figure 4 in the new Manuscript).

Q9. Lines 249-252. The authors use the ionic current simulations to identify the region of the pore that modulates the current most. In doing so, they conclude that “ion flux was clearly limited around select residues”. Well, the ionic current through any cross-section of the channel must be constant to satisfy the current continuity requirement. The only situation in which the current through different sections of the channel can be different is if the

simulation has not yet reached a steady state regime. In that case, the simulation outcome will depend on initial conditions.

A9. We agree with the reviewer and we have rephrased our explanation that was originally misleading. We have made clearer that monitoring ion distribution within the pore, we observed that ions are also spatially confined at the 2 narrow constrictions corresponding with the sensing spots (Page 8): *“Looking at the distribution of ions along the pore during MD simulations with dA_{14} (Figure 4d), we observed large ion occupancy in between the two constriction points, while ions were more confined around R220/R282 and K238/K242, likely contributing to determine the specific ion current features of the aerolysin pore.”*

Q10. Line 311: The claim of “de facto” better sensitivity must be supported by signal to noise analysis.

A10. As the open pore current of aerolysin is clearly smaller than that of other pores, this results in smaller current differences. What we wanted to stress actually was that the narrower constriction points allow aerolysin to detect smaller oligonucleotides. We have rephrased this point in the manuscript (Page 10): *“In comparison to α -hemolysin and MspA, the sensing spot produced by a heptameric ring of large arginine residues is significantly narrower (less than 0.9 nm), allowing aerolysin to detect oligonucleotides as short as 2 bases, which is not feasible for the two other pores.”*

Q11. Line 355: A 5' phosphate can be added to DNA using a standard CHARMM patch, there is no need to do parameterization.

A11. We actually overlooked this and we now have run the new MD simulations using the standard CHARMM patch. We also checked if previous MD simulations were affected by our parameterization by comparing with MD runs using the CHARMM patch and found similar behaviour and no relevant differences.

Q12. The authors appear to miss the following two critical MD studies that examined the effect of arginine mutants on DNA transport through MspA [ACS Nano, 6:6960] and determined the molecular origin of the sequence specific ionic current blockades [ACS Nano 10: 4644]. The authors are asked to discuss the result of their work in the context of these studies.

A12. We have now included a comparison with those works on the manuscript: *“This observation agrees with previous MD studies using the MspA pore, which revealed that the inclusion of extra positive side chains in the lumen resulted in an increase in translocation time²⁶.”* (Page 3). *“Interestingly, during MD, transient π -stacking interactions were observed around both constriction points, between the 11th and the 12th bases, or the 3th and 4th. Similar interactions have been previously described as related to ionic current blockade in the MspA pore³³.”* (Page 7) *“Our results show a good relationship between ion flux decrease and the pore diameter, suggesting that the main determinant for the residual current is produced by pore steric hindrance, as already proposed for other nanopores^{33, 40, 41}”* (Page 10)

Q13. Line 369. The equilibration must be performed in the NPT, not NVT ensemble. NPT ensemble is required for the system to acquire equilibrium dimensions and to reach the target pressure of 1 bar.

A13. Actually, this was indeed a typo, as usual equilibration and production were run in the NPT ensemble. We have now corrected this point in the methods section.

Q14. Line 370: Specify the spring constant of the restraints and duration of each equilibration step.

A14. The spring constant and the duration of each equilibration step have been included on the methods section.

Q15. Line 374: Using SMD to introduce a DNA strand into biological nanopore is not the best idea. The application of the force to the leading nucleotide will stretch the DNA strand, which could lead to artificially elongated DNA conformation in the nanopore. A much more superior approach is in using a Grid-Steered MD, see [JCP 127:125101] for details. In any event the amount of stretching must be detailed in SI.

A15. We agree that *Grid-Steered MD* is superior to SMD. However, in our hands even at high voltages, R220 blocks DNA translocation, avoiding crossing the constriction point, possibly due to the lack of a vestibule in aerolysin, together with its narrower lumen compared to other pores. Thus, the only way we managed to translocate DNA was by using a pulling force of minimum $100 \text{ kJ mol}^{-1} \text{ nm}^{-2}$. As requested, we have now calculated the stretching of DNA during SMD, and its relaxation along MD (see Figure S21), measured as the length between the first and last phosphate groups, following a metric already published by others³. As for the plots, there is a full relaxation of the DNA stretching produced by SMD already in the first ns of unbiased MD simulations, thus that it reaches stabilization of the initial configuration with a length of 73 Å, lower than the estimated contour length for dA₁₄, being that inter-phosphate distances of ssDNA can range between 5.9 Å and 7 Å depending on the sugar pucker⁴⁻⁵. A mention to this stretching measure has been included on the methods section on Page 12: “*The stretching events during SMD, and its relaxation during MD, have been measured based on the change of the interstrand distance, as already performed by others*⁴⁹ (Supplementary Figure S21).”

Q16. The authors carried out several simulations of the system at different voltages but they never report the actual simulated current. What was the open pore and blockade currents obtained using the MD methods? Please add integrated current plots to the SI.

A16. The objective of this manuscript was getting the first insights into the main aerolysin sensing spots, and therefore we had not originally performed calculations of ionic current. Following your request, we have done it now. The current was calculated following two methods already described by others⁶⁻⁷. These 2 methods were applied similarly to two systems, one with dA₁₄ fully inserted in the pore, and the other one with no DNA.

In the first method, the current is estimated by counting the number of ions that cross the channel (N_{K^+} and N_{Cl^-}), according to the following formula, where q is the charge of the ion, and Δt is the time interval of the measurement:

$$i = (N_{K^+} + N_{Cl^-})q/\Delta t$$

According to this formula, the average current for the free-DNA system was $\approx 127.5 \text{ pA}$ under 250mV, while for the DNA-bound aerolysin, the current was nearly zero, as no ion completely crossed the pore. Longer simulation times would however be required to have more robust estimation of ion current by using this method.

The second method takes the motion of all ions into account, estimating the current with the following formula, where z_i and q_i are the z coordinate and the charge of atom i , respectively:

$$I(t) = \frac{1}{\Delta t L_z} \sum_{i=1}^N q_i [z_i(t + \Delta t) - z_i(t)]$$

Only results using this second method are presented on the manuscript, on the methods section (Page 13) and in the Supplementary Figure S22. Briefly, the current has average values of 120 and 0 pA in the systems without/with DNA, respectively, both at 250 mV. These values are consistent with the calculations performed with the first method and are importantly in line with the experimental values measured for the open pore (125 pA) and DNA-bound aerolysin pore (5 pA).

Q17. Was the protein structure restrained during the E-field simulation?

A17. We observed that there was no need for additional restraints at the current used in this work, the aerolysin structure remained stable along the simulation. However, when we tried higher voltages (we are not including the results here because they were not relevant for this study) we actually needed to apply restraints on the α -carbons.

Response to the Reviewer #2

Thank you very much for your constructive suggestions and comments, which have been very helpful toward improving our manuscript. Accordingly, we have put a significant effort to revise and improve our manuscript on these three aspects: (i) revising figures; (ii) complementing the study with other DNA fragments (including dA₁₀, dA₁₅ and dA₂₀) from both experiments and simulations; (iii) including some important references to better explain our results.

The revised parts are marked in red-colour font in the main text. A detailed, point-by-point response is reported here below:

Q1. In Figure 3, dA14 was used for single-abase screening. Indeed, the current amplitude for dA14 translocation can be changed with the abase position. However, it is not convincing to utilize the current change to map the sensing spot. Each current block observed represents an entire procedure for DNA dynamic translocation, which involves every base that enters, slides and leaves the pore. This process must include many conformations for DNA in different positions in the pore (including partial entering the pore). That is to say, Figure 3b and c shows only one conformation that has a dA14 fully trapped in the pore during translocation. The experiments in the paper lack convincing evidence to justify how this current is generated, and whether the current amplitude for translocation is generated by this illustrated conformation or other conformations. Therefore, it is impossible to map the sensing spots only based on the abase DNA current amplitude.

A1: Thank you very much for having raised this point. Yes, it is true that each current blockade we observed represents a complex process of DNA translocation. As illustrated in the main text and SI, the averaged levels of each blockade current show a Gaussian distribution, which means that each oligonucleotide produced a specific blockade value within this distribution. This specific value would therefore correspond to a preferential conformation. Importantly, we have shown that dA₁₄ blocked the pore presenting much longer translocation time than either longer or shorter oligonucleotides (see Figure 1 and Q2/A2 here below). This peculiar blockage is interpreted by MD as dA₁₄ filling entirely the pore lumen. Thus, the assumption that dA₁₄ as well as single dA₁₄ abase constructs retain this conformation in the pore provided us the working hypothesis for mapping the sensing spots at single-nucleobase resolution, by using single-abases nucleotides at different positions. A posteriori, our results are consistent with the structural and electrostatic features of the pore, thus that it appears reasonable to propose the existence of 2 main constrictions. Moreover, this same protocol has been already used for α -hemolysin and MspA pores, where single-abase with streptavidin/dsDNA were used to stabilize the oligonucleotide inside the pore and gain additional molecular information about pore sensitivity⁸⁻¹⁴.

Q2: In Figure 2, dA14 was used to study the translocation direction. By attaching a streptavidin to block one end, dA14 can only enter the pore from the other end. Both translocation directions can be identified by two current amplitudes in the histogram (Figure 2a). However, the current distributions for all other DNAs (dA10 to dA20) in Figure 1c and d only show a single current component. This controversy causes confusion, which one is correct? Can all DNAs translocate in two directions? Furthermore, can dA14 abase variants translocate in both directions? Please verify experimentally. If they do, there should be two set of I/I₀ data for each abase variant that should be shown in Figure 3b to map the pore amino acids. Also, the component assignments in histograms in Figure 4 would be more complicated for base discrimination.

A2: We have clarified this point in the revised manuscript. First, it should be noted that the experimental data showed in Figure 1 and Figure 3 are obtained in 1M KCl, while the data of

Figure 2 and Figure 4 are obtained in 3.0 M KCl. According to the scatter plot of Figure 1c, it is clear that all the oligonucleotides only have one distribution at 1M KCl conditions, which means most of the oligonucleotides only have one preferential direction during the translocation at 1M. To better understand in which direction oligonucleotides were captured and transported, we investigated the capturing process using MD simulations. As illustrated in Figure 2b and Figure S4, MD simulations predicted that 3' end is easier to be captured since the force needed for translocation by the 5' end was higher. This extra resistance is likely originated by the enantiomeric properties of the desoxy-D-ribose, which produces more steric hindrance for DNA bases as they move down the channel encountering large side chains.

Interestingly, we found that at higher salt solution (e.g. 3M KCl), there are two peaks (PI and PII) of current distribution as showed in Figure 2a (Top), but the percentage of second peak was around 20% at 100 mV. Considering the possibility that oligonucleotides can be also captured by the 5' side, we further used the streptavidin-immobilized method to unambiguously assign the two peaks at 3M KCl (Figure 2a) and thus be able to interpret the results at lower concentration. According to the results, we know that PI with a lower I/I_0 was induced by the ssDNA entering from 3' end while PII with a higher I/I_0 induced by the entering from 5' end. In Figure 4, we used the PI to identify the different kinds of oligomers.

In conclusion, the main result is that at low salt concentration ssDNA is captured at the 3' end, while the 5' end capture is very difficult, but the energy barrier associated with this process can be reduced under a higher voltage or higher salt concentration. We have revised the manuscript (at Page 5) to explain this point that was unclear in the original version. *“The scatter plots of **Figure 1c** show a unique direction of entry for DNA. In order to understand which was this preferential direction, we used a higher salt concentration (3M), which produced two well defined peaks in the I/I_0 histograms of dA_{14} , each corresponding to one different entry direction (**Figure 2a**).”*

Q3: dA_{14} is a special DNA. According to the model shown in the paper, it is about the same length as the aerolysin pore, 10 nm. Also its block duration is the longest as shown in Figure 2e. It was interpreted that “the interaction between dA_{14} and the aerolysin nanopore is stronger, likely due to the optimal spatial filling of dA_{14} within the ~10 nm long pore channel,” and “This optimal steric and electrostatic match 119 of dA_{14} within the lumen is likely at the origin of the observed increased duration of 120 translocation time.” While this explanation is acceptable, why do longer DNAs (dA_{15} ~ dA_{20}) that contain the same dA_{14} domain and experience the same interaction in the pore have much shorter translocation duration? Can longer dA_{15} ~ dA_{20} get the similar abase screening result and single base sensitivity as dA_{14} . Can the conclusion drawn from dA_{14} such as single base sensitivity and sensing spots be generalized to long DNAs? This has not been evidenced yet and limited the impact.

A3: Thank you very much for your question. To explain this better, we have complemented the revised manuscript with MD simulations of dA_{20} and dA_{10} , following the same procedure used for dA_{14} . We performed MD for ssDNA with 3' and 5' end orientation with the pore exit.

For the longer dA_{20} , the results showed that this oligonucleotide is in fact longer than the pore length, resulting, depending on starting conditions, in the accumulation of the remaining bases at the pore entry or exit (see Figure S2). For dA_{10} , we observed as expected that it does not completely fill the pore, and the DNA tends to accumulate at the pore exit. The sampling of these MD simulations does not allow us to draw any definitive quantitative description of the translocation process (as here they are mostly used as an exploratory means to interpret and guide experiments), however, these additional conformations are suggesting that dA_{14} is trapped longer in the pore because its optimal match with the full length of the channel, which might block for longer ions translocation. This is not the case for shorter oligonucleotides, that have already the tendency to exit the pore, and for longer,

which having a portion always exposed to the bulk that can in turn facilitate ions translocation and accelerate the exit of the ssDNA from the pore. A discussion regarding this aspect has been added to the revised manuscript (see Pages 3-4 and Supplementary Figure S2): “This observation agrees with previous MD studies using the MspA pore, which revealed that the inclusion of extra positive side chains in the lumen resulted in an increase in translocation time²⁶. The optimal steric and electrostatic match of dA₁₄ within the lumen is likely at the origin of the observed increased duration of translocation time. By comparison, shorter (dA₁₀) and longer (dA₂₀) oligonucleotides did not show the same perfect match during MD simulations (**Supplementary Figure S2**). For the longer dA₂₀, the results showed that this oligonucleotide remains in fact more extended than the pore length, resulting, depending on starting conditions, in the accumulation of the remaining bases at the pore entry or exit. On the other hand, the shorter dA₁₀ does not completely fill the pore, and the DNA tends to accumulate at the pore exit. Contrary to the optimal match of dA₁₄, shorter or longer oligonucleotides may accelerate their exit from the pore by facilitating a faster restoration of ion current thanks to conformations that allow for a partially filled pore lumen (e.g., dA₁₀) or solvated bases in the bulk (e.g., dA₂₀).”

In addition, we tried single-channel experiments for longer DNAs, such as dA₁₅ and dA₂₀. For example, we designed only one nucleobase variation at the 11th position of the dA₁₅. Then, we mixed and added four oligomers in the *cis* chamber of the aerolysin nanopore. As illustrated in Figure R2 (review only), we can identify four oligomers but have an overlap between dA₁₅x11-T and dA₁₅x11-C, not as perfect as for the dA₁₄x11 series. Therefore, we think that it is possible to extend our conclusion to longer ssDNA, but the separation will be reduced with an increasing length of oligomer since we used the mean blockade current to do the separation. To further study this, we could (i) introduce an enzyme to control the ssDNA translocation in the aerolysin system; (ii) decrease the length of the pore; or (iii) improve software to achieve a step by step current readout, but all these are out of the scope of the present work, but will be likely pursued in the future. Our manuscript provides basic information about the sensing properties of aerolysin for single molecule reading. With this information, we will be able to engineer the pore for specific applications.

Figure R2. Discrimination of all 4 DNA bases by dA₁₅X11 sites of the aerolysin nanopore. Histograms of residual current caused by the four kinds of nucleobase in a dA₁₅ strand. The sequence information of four ssDNA was illustrated in the top of the figure. The data were obtained in 3.0 M KCl, 10 mM Tris, and 1.0 mM EDTA, pH=8.0, 17 ± 2 °C at the bias potential of + 100 mV. The number of blockades is 3,000.

Q4. Figure 1b needs to show the entire block events, rather than the initial parts, for all polydA DNAs, such that the current profile of the blocks can be analysed.

A4: Thanks for your suggestion. We have revised the representation of typical events for all polydA DNAs in the new Figure 1b.

Q5. Figure 4a, b and c should use I/I_0 to represent the current blocks in x-axis, to keep consistence with and compare with current amplitude in other figures.

A5: To keep consistence, we have changed the residual current to I/I_0 in x-axis in Figure 4a, b and c.

References

1. Zhang, J.; Shklovskii, B. I., Effective charge and free energy of DNA inside an ion channel. *Physical Review E* **2007**, *75* (2), 021906.
2. Wang, Y.; Tian, K.; Du, X.; Shi, R.-C.; Gu, L.-Q., Remote Activation of a Nanopore for High-Performance Genetic Detection Using a pH Taxis-Mimicking Mechanism. *Anal. Chem.* **2017**, *89* (24), 13039-13043.
3. Marin-Gonzalez, A.; Vilhena, J. G.; Perez, R.; Moreno-Herrero, F., Understanding the mechanical response of double-stranded DNA and RNA under constant stretching forces using all-atom molecular dynamics. *Proc. Natl. Acad. Sci. USA* **2017**, *114* (27), 7049-7054.
4. Olson, W. K., Configurational Statistics of Polynucleotide Chains. A Single Virtual Bond Treatment. *Macromolecules* **1975**, *8* (3), 272-275.
5. Saenger, W., *Principles of Nucleic Acid Structure*. **1984**, Springer-Verlag New York.
6. Aksimentiev, A.; Schulten, K., Imaging α -Hemolysin with Molecular Dynamics: Ionic Conductance, Osmotic Permeability, and the Electrostatic Potential Map. *Biophys. J.* **2005**, *88* (6), 3745-3761.
7. Balijepalli, A.; Robertson, J. W. F.; Reiner, J. E.; Kasianowicz, J. J.; Pastor, R. W., Theory of Polymer–Nanopore Interactions Refined Using Molecular Dynamics Simulations. *J. Am. Chem. Soc.* **2013**, *135* (18), 7064-7072.
8. Stoddart, D.; Heron, A. J.; Mikhailova, E.; Maglia, G.; Bayley, H., Single-nucleotide discrimination in immobilized DNA oligonucleotides with a biological nanopore. *Proc. Natl. Acad. Sci. USA* **2009**, *106* (19), 7702-7707.
9. Stoddart, D.; Heron, A. J.; Klingelhoefer, J.; Mikhailova, E.; Maglia, G.; Bayley, H., Nucleobase Recognition in ssDNA at the Central Constriction of the α -Hemolysin Pore. *Nano Lett.* **2010**, *10* (9), 3633-3637.
10. Stoddart, D.; Maglia, G.; Mikhailova, E.; Heron, A. J.; Bayley, H., Multiple Base-Recognition Sites in a Biological Nanopore: Two Heads are Better than One. *Angew. Chem. Int. Ed.* **2010**, *49* (3), 556-559.
11. Wallace, E. V. B.; Stoddart, D.; Heron, A. J.; Mikhailova, E.; Maglia, G.; Donohoe, T. J.; Bayley, H., Identification of epigenetic DNA modifications with a protein nanopore. *Chem. Commun.* **2010**, *46* (43), 8195-8197.

12. Jin, Q.; Fleming, A. M.; Johnson, R. P.; Ding, Y.; Burrows, C. J.; White, H. S., Base-Excision Repair Activity of Uracil-DNA Glycosylase Monitored Using the Latch Zone of α -Hemolysin. *J. Am. Chem. Soc.* **2013**, *135* (51), 19347-19353.
13. Johnson, R. P.; Fleming, A. M.; Beuth, L. R.; Burrows, C. J.; White, H. S., Base Flipping within the α -Hemolysin Latch Allows Single-Molecule Identification of Mismatches in DNA. *J. Am. Chem. Soc.* **2016**, *138* (2), 594-603.
14. Tan, C. S.; Riedl, J.; Fleming, A. M.; Burrows, C. J.; White, H. S., Kinetics of T3-DNA Ligase-Catalyzed Phosphodiester Bond Formation Measured Using the α -Hemolysin Nanopore. *ACS Nano* **2016**, *10* (12), 11127-11135.

Reviewers' comments:

Reviewer #1 (Remarks to the Author):

The authors have made considerable improvements to the manuscript, addressing most of the questions raised in the previous round of review. Unfortunately, the revised version of the manuscript still lacks essential information about the procedures used in MD simulations. Otherwise, the manuscript describes an important study that will for sure be of great interest to the nanopore research field.

First, this reviewer would like to note that the authors did not provide their responses to the general points listed at the beginning of the referee report, in particular to point 1, which was:

--- 1. It is not clear whether the single nucleotide discrimination is possible for DNA fragments that are not exactly 14 nucleotide long. The fact that the authors were able to observe clear ionic current signatures from individual nucleotides suggest that the time the DNA strand spends entering and leaving the nanopore is negligible compared to the residence of a full 14-nucleotide fragment. So what would happen to a DNA that is, say 16 nucleotides long?

This specific concern was about the experimental part of the work (not MD simulations), but perhaps it was clearly formulated. Fortunately, reviewer 2 expressed a similar concern (Q3), which the authors addressed adequately in their response.

The authors are now asked to revise the manuscript to provide the missing information about the MD simulation protocols. This information is essential for other groups to reproduce the simulation results reported in the manuscript.

Q1. Simulation setup.

- Specify the size of the membrane patch added during the model building process
- Explicitly name the residues of the protein that were omitted in the truncated pore model.
- Specify the type and concentration of the electrolyte.
- Specify the approximate dimension of the systems (in x, y and z) at the end of the equilibration simulation.
- Add to SI information a molecular graphics image of the final equilibrated system that illustrates the location of the protein with respect to the lipid bilayer membrane and the extent of the solvent volume. The image can be combined with RMSD and Z-dimension plots (see Q3 and Q4) into one SI figure.

Q2. Again, how was the protein restrained during the simulations? Lines 404-407 say " While the lipid restraints gradually decreased to zero, the protein and DNA were kept fixed. Afterwards, it was simulated for 150 ns in the NPT ensemble, with the Nose-Hoover thermostat for temperature coupling and the Parrinello-Rahman for semi-isotropic pressure coupling, gradually releasing the protein and DNA restraints". Unfortunately, this is not specific enough to be reproduced by another group.

- Specify duration of each step of the restrained simulation; this was requested in the previous round of review

- Specify the rate of restrain removal in the last step, and the duration of free equilibration, if any was performed. If the system was never simulated without any restraints, please revise the method section so that it does not leave such an impression.

Q3: The authors simulated a truncated model of the channel, an approach used previously by several groups [ACS Nano, 6:6960 (2012), and Refs 26, 36 and 44]. In all those previous studies,

the truncated protein was restrained to crystallographic coordinates to prevent structure deterioration. According to the authors' response to Q17 and lines 404-407 of the revised manuscript, it may appear that the authors did not use any restraints during the applied electric field simulations. If so, did the protein maintain its structure?

-- Specify whether restraints were used during electric field simulations

-- In the SI information, provide a plot of backbone RMSD for the equilibration and applied electric field simulations from the initial pseudo atomic model.

Q4. This reviewer is confused about the ensemble used for the applied electric field simulations. According to the authors' response (Q13) it was NPT, but it should have been NVT. Using NPT and applied electric field can affect pressure control, resulting in stretching of the simulation system along the direction of the applied electric field. Either this is another typo in the author's response or the authors indeed combined NPT and applied electric field.

--- In the former case, specify the ensemble in the methods section.

--- In the latter case, show that the system maintained stable dimensions by plotting the system's dimensions along the z axis during ionic current simulation. This can be combined with an RMSD plot requested in Q3.

Q5: Now the authors report the simulated ionic current which is in great quantitative agreement with experiment. This is a great result, but it rises additional questions. The force field used by the authors is known to overestimate bulk electrolyte conductivity (by a factor of 1.6 for 1M KCl), so comparing experimental and simulation conductance of a biological nanopore usually requires scaling the simulated conductance by the ratio of experimental and simulated bulk conductivities, see SI of Ref 33 for details. Furthermore, pore truncation should have also increased the open pore conductance with respect to the unmodified nanopore (see SI of Ref 33 again). It is therefore somewhat surprising that the open pore current agrees with experimental value so well. How do you explain that?

Q6. This reviewer is puzzled by the authors' choice of Ref 26 as a sole reference to previous MD simulation work on MspA. ACS Nano, 6:6960 (2012) predates Ref. 26 by 3 years and describes the first comprehensive MD study of the MspA system, including the effect of arginine mutations. It was also that study that described the pi-stacking interactions between DNA backbone and arginine residues of the pore. Please reconsider your choice.

Reviewer #2 (Remarks to the Author):

I have read the revised report and authors' reply to my questions. Thank you.

The main point of this work is utilizing a 14 bases poly A DNA to identify the sensing spots in aerolysin for single base detection. Each nucleotide of poly A was replaced with an abase motif, then each poly A carrying a single abase electrically transported through the aerolysin pore. The resulting change in the translocation current due to the abase substitution was measured to map the supposed sensing spots.

I also read other published works on detecting sensing spots or single base variation in protein pores. These papers have been shown by the authors in the rebuttal letter and listed below. The

DNA in all these studies was immobilized in the nanopore by either attachment with a streptavidin (mainly by the Bayley group) or using a double strand DNA (White and Burrow groups). The advantage of using "immobilized" DNA is that each base is precisely fixed at a specific position in the nanopore. In other words, the base position is known. Only in this immobilized conformation, the current variation by each base change can be correlated with a local sensing spot.

In contrast to measuring fixed DNAs, this report measures the DNA translocation current. The problem is that the entire translocation procedure involves many conformations, and the abase position in the pore continuously moves along with translocation. The resulting nanopore current is the average of all translocation steps, rather than corresponding to a specific abase position. Without fixing the abase position, it is impossible to map the abase interaction with the specific sensing spot. The authors explained that "the averaged levels of each blockade current show a Gaussian distribution, which means that each oligonucleotide produced a specific blockade value within this distribution. This specific value would therefore correspond to a preferential conformation." However, this seems to be an assumption. There has been no direct evidence showing the poly A used in the report is somewhat temporarily fixed in this "preferential conformation".

As analyzed above, the revised report still lacks substantial improvement on this concern. For example, there is no result and analysis for mapping the sensing spots with immobilized DNA, such as using streptavidin tag, as in many published works. Currently, It remains difficult to agree with the conclusion that the sensing spots in aerolysin can be accurately determined by measuring the change in the abase DNA translocation current.

References in authors' rebuttal letter:

8. Stoddart, D.; Heron, A. J.; Mikhailova, E.; Maglia, G.; Bayley, H., Single-nucleotide discrimination in immobilized DNA oligonucleotides with a biological nanopore. *Proc. Natl. Acad. Sci. USA* 2009, 106 (19), 7702-7707.
9. Stoddart, D.; Heron, A. J.; Klingelhoefer, J.; Mikhailova, E.; Maglia, G.; Bayley, H., Nucleobase Recognition in ssDNA at the Central Constriction of the α -Hemolysin Pore. *Nano Lett.* 2010, 10 (9), 3633-3637.
10. Stoddart, D.; Maglia, G.; Mikhailova, E.; Heron, A. J.; Bayley, H., Multiple Base-Recognition Sites in a Biological Nanopore: Two Heads are Better than One. *Angew. Chem. Int. Ed.* 2010, 49 (3), 556-559.
11. Wallace, E. V. B.; Stoddart, D.; Heron, A. J.; Mikhailova, E.; Maglia, G.; Donohoe, T. J.; Bayley, H., Identification of epigenetic DNA modifications with a protein nanopore. *Chem. Commun.* 2010, 46 (43), 8195-8197.
12. Jin, Q.; Fleming, A. M.; Johnson, R. P.; Ding, Y.; Burrows, C. J.; White, H. S., Base-Excision Repair Activity of Uracil-DNA Glycosylase Monitored Using the Latch Zone of α -Hemolysin. *J. Am. Chem. Soc.* 2013, 135 (51), 19347-19353.
13. Johnson, R. P.; Fleming, A. M.; Beuth, L. R.; Burrows, C. J.; White, H. S., Base Flipping within the α -Hemolysin Latch Allows Single-Molecule Identification of Mismatches in DNA. *J. Am. Chem. Soc.* 2016, 138 (2), 594-603.
14. Tan, C. S.; Riedl, J.; Fleming, A. M.; Burrows, C. J.; White, H. S., Kinetics of T3-DNA Ligase-Catalyzed Phosphodiester Bond Formation Measured Using the α -Hemolysin Nanopore. *ACS Nano* 2016, 10 (12), 11127-11135.

Response to Reviewers

We thank the reviewers for their assessment of our manuscript and respond here below point-by-point to their concerns. All the revised parts of the manuscript and supporting information are marked in blue.

Reviewer 1

Q1. Simulation setup.

- Specify the size of the membrane patch added during the model building process
- Explicitly name the residues of the protein that were omitted in the truncated pore model.
- Specify the type and concentration of the electrolyte.
- Specify the approximate dimension of the systems (in x, y and z) at the end of the equilibration simulation.
- Add to SI information a molecular graphics image of the final equilibrated system that illustrates the location of the protein with respect to the lipid bilayer membrane and the extent of the solvent volume. The image can be combined with RMSD and Z-dimension plots (see Q3 and Q4) into one SI figure.

A1: **The following additions were made:**

- *The size of the membrane patch has been specified (Page 12): "A ~11x11 nm² membrane bilayer was modeled by 1-palmitoyl-2-oleoylphosphatidylcholine (POPC) lipids using the charmm-gui server,"*
- *The residues that were omitted in the model have been specified (Page 12): "The aerolysin pore was described using the equilibrated conformation defined by previous MD simulations of the aerolysin pore performed by some of us⁴⁵. This original model was reduced by removing the membrane binding domains (i.e., residues 24-195, 301-408 and 425-447), and the last strand (residues 409-424) was linked to the previous one by creating a loop between amino acids A300 and Q409 using the software Modeller v9.13⁴⁶. The simulations were thus performed on the membrane spanning β -barrel only, which remained stable as the full pore unit during all the MD simulations presented here. This same approximation has been already used for translocation studies in other PFTs^{26, 47}.*
- *The type and concentration of the electrolyte have been specified (Page 12): "This system was then solvated in a 1 M KCl water box with initial dimensions ~11x11x21.7 nm³.*
- *The dimensions at the end of the equilibration had the following average values: 10.6, 10.6 and 21.3, for x, y and z, respectively. We added the final box size values on the manuscript (Page 12): "The approximate dimensions of the box at the end of the equilibration were ~10.6x10.6x21.3 nm³"*
- *An image showing the location of the protein with respect to the lipid bilayer membrane and the extent of the solvent volume at the end of the equilibration has been added in Supplementary Figure S23." (Page 13).*

Q2. Again, how was the protein restrained during the simulations? Lines 404-407 say “While the lipid restraints gradually decreased to zero, the protein and DNA were kept fixed. Afterwards, it was simulated for 150 ns in the NPT ensemble, with the Nose-Hoover thermostat for temperature coupling and the Parrinello-Rahman for semi-isotropic pressure coupling, gradually releasing the protein and DNA restraints”. Unfortunately, this is not specific enough to be reproduced by another group.

-- Specify duration of each step of the restrained simulation; this was requested in the previous round of review

-- Specify the rate of restrain removal in the last step, and the duration of free equilibration, if any was performed. If the system was never simulated without any restraints, please revise the method section so that it does not leave such an impression.

*A2. We have corrected this part (Page 12) and introduced a better description of the equilibration protocol as **Supplementary Table S3**, including the duration of each step and the rate of restrain removal.*

*“The system was first minimized using the steepest descent algorithm, and afterwards equilibrated using a similar protocol of that suggested by the charm-gui server⁵³, but in 2 blocks of 6 steps each, totalizing 975 ps. In the first block of equilibration, the lipid restraints were gradually reduced to zero, while those on protein and DNA were maintained, and were only gradually and completely removed in the second block. Afterwards it was simulated for 10 ns without restraints. A more detailed description of this equilibration protocol can be found as **Supplementary Table S3**. ”*

Q3: The authors simulated a truncated model of the channel, an approach used previously by several groups [ACS Nano, 6:6960 (2012), and Refs 26, 36 and 44]. In all those previous studies, the truncated protein was restrained to crystallographic coordinates to prevent structure deterioration. According to the authors’ response to Q17 and lines 404-407 of the revised manuscript, it may appear that the authors did not use any restraints during the applied electric field simulations. If so, did the protein maintain its structure?

-- Specify whether restraints were used during electric field simulations

-- In the SI information, provide a plot of backbone RMSD for the equilibration and applied electric field simulations from the initial pseudo atomic model.

A3: Yes, the protein did maintain its final structure without restraints. As we formulated in a previous article by some of us², we think that the double beta barrel present at the top of the pore is responsible for the prion-like stability of the aerolysin pore. We think that this fold is keeping our model system stable at high electrical fields, as we observe no modification of the structure. Only when very strong fields were applied (not presented in this work) we needed to apply restraints to keep the lower transmembrane region restrained. However, the

truncated region, on the top of the barrel, was still not modifying its structure at those high electric fields.

-- We included a comment on the method section to explain that no restraints were used (Page 13): *"The protein remained very stable during simulations at this electric field without the need of applying position restraints. This is likely due to the prion-like stability conferred to the pore by the concentric double beta-barrel conformation at the extracellular end (see **Supplementary Figure S23** for RMSD plots of the equilibration and MD simulation at applied voltage).*

-- The plots have been included as **Supplementary Figure S23**.

Q4. This reviewer is confused about the ensemble used for the applied electric field simulations. According to the authors' response (Q13) it was NPT, but it should have been NVT. Using NPT and applied electric field can affect pressure control, resulting in stretching of the simulation system along the direction of the applied electric field. Either this is another typo in the author's response or the authors indeed combined NPT and applied electric field.

--- In the former case, specify the ensemble in the methods section.

--- In the latter case, show that the system maintained stable dimensions by plotting the system's dimensions along the z axis during ionic current simulation. This can be combined with an RMSD plot requested in Q3.

A4: We apologize we didn't explained this part better previously. We actually used/tested 2 approaches while setting up our simulations: (i) NPT ensemble combined with the electric field, but fixing the Z-axis dimension to avoid stretching. We believe that this approximation can represent a more realistic model, since it allows the membrane to relax on the (x,y) plane. (ii) We also performed the same simulations in NVT ensemble combined with the electric field (as the reviewer was expecting) and observed practically identical results concerning box dimensions, as well as DNA positioning and interactions. We have also calculated the ion conductivity in both cases and results were always consistent.

-- It has been specified on the methods section the use of NPT ensemble with z-axis dimension fixed in all simulations after the equilibration (Page 12): *"For all further simulations (SMD and MD of selected snapshots), we chose an integration step of 2 fs. A temperature of 22 °C was controlled with the Nose-Hoover thermostat and the Parrinello-Rahman method was used for semi-isotropic pressure coupling."* And (Page 13): *"In all simulations with applied electric field, the dimension of the box along the z-axis was kept fixed."*

-- Thus the z-axis dimension is constant value during the whole simulation, and we decided not to include any plot.

Q5: Now the authors report the simulated ionic current which is in great quantitative agreement with experiment. This is a great result, but it rises additional questions. The force field used by the authors is known to overestimate bulk electrolyte conductivity (by a factor of 1.6 for 1M KCl), so comparing experimental and simulation conductance of a biological nanopore usually requires scaling the simulated conductance by the ratio of experimental and simulated bulk conductivities, see SI of Ref 33 for details. Furthermore, pore truncation should have also increased the open pore conductance with respect to the unmodified nanopore (see SI of Ref 33 again). It is therefore somewhat surprising that the open pore current agrees with experimental value so well. How do you explain that?

A5: Contrary to alpha-hemolysin or MspA, the aerolysin pore has no vestibule. The truncated extracellular regions extend surrounding the pore, away from the pore entry, separated from it by the double beta barrel region. We therefore expect these regions, which we truncated, are not affecting substantially the ion conductance. They may be involved in aerolysin association to the membrane and/or in the pore formation process, as proposed in a previous work by some of us¹. Moreover, in the revision we now discuss how our MD approach is overestimating the calculated absolute current values. Based on the works cited therein, we found that bulk electrolyte conductivity can be overestimated from 10 to 40% using the CHARMM force field with different simulation parameters conditions. As it was not our scope in this manuscript to have a rigorous estimation of the current, we have discussed how our calculation has to be considered overestimated within this range (Pages 8 and 14): “The ionic current was calculated using a well-established method^{33, 36} for an aerolysin pore system without ssDNA and for the dA₁₄ system giving results of 118.6 ± 1.4 and 0.7 ± 0.3 pA, respectively. Our approach, however, is known to overestimate the absolute current mainly due the overestimation by the CHARMM force field of the bulk electrolyte conductivity by ~10-40%^{33, 36, 57, 58}. Given the approximate nature of our approach, the calculated current is reasonably well in line with the experimental values (125.0 and 5.0 pA) (Supplementary Figure S20).”

Q6. This reviewer is puzzled by the authors' choice of Ref 26 as a sole reference to previous MD simulation work on MspA. ACS Nano, 6:6960 (2012) predates Ref. 26 by 3 years and describes the first comprehensive MD study of the MspA system, including the effect of arginine mutations. It was also that study that described the pi-stacking interactions between DNA backbone and arginine residues of the pore. Please reconsider your choice.

A6: There was a mistake in the reference 26 in fact. We apologise for that and we have corrected the mistake, substituting the wrong reference by ACS Nano, 6:6960 (2012).

References

1. Cirauqui, N.; Abriata, L. A.; van der Goot, F. G.; Dal Peraro, M., Structural, physicochemical and dynamic features conserved within the aerolysin pore-forming toxin family. *Sci. Rep.* **2017**, 7 (1), 13932.
2. Iacovache, I.; De Carlo, S.; Cirauqui, N.; Dal Peraro, M.; van der Goot, F. G.; Zuber, B., Cryo-EM structure of aerolysin variants reveals a novel protein fold and the pore-formation process. *Nat. Commun.* **2016**, 7, 12062.

Response to reviewer #2

We understand the concerns of this reviewer and we would like to explain our position on this point more extensively. We are aware that other methods are available to detect sensing spots on biological pores, and immobilized DNA constructs have been used in the past with success. Nonetheless, in this work we chose not to use this strategy for a series of reasons that have to do with the peculiar nature of aerolysin. We think that any given method is not necessarily the best option for all systems.

Based on our experience working with aerolysin, we think that its unique structure provides the possibility to map the sensing spots using free DNA rather than a complicated immobilized DNA system (e.g. by fusion with streptavidin). If on one side immobilized DNA is surely trapped in the pore, it is also perturbing the whole system in ways that we cannot control. For instance, by using streptavidin the attached DNA have to be biotinylated at the 5'-side, and biotin can produce a larger flexibility of the whole system since its structure is more flexible than DNA. However, the main reason why other authors were using streptavidin to immobilize DNA is because they were working with proteins forming broader and shorter pores than aerolysin and, therefore, the DNA translocation was too fast to detect anything.

Therefore, having the possibility to work with free DNA is in principle preferable if one can achieve similar conditions. We found out that aerolysin is capable to trap A14 oligonucleotides constructs for quite long times (11.91 ± 0.29 ms), much longer than other shorter or longer constructs, and much longer than any other pore. This first observation allowed us to use A14 as a *quasi-immobilized probe*, which for aerolysin can substitute a DNA-streptavidin system. To prove and rationalize this point we also performed all-atom MD simulations that gave evidences that the reason of such a long translocation time is likely due to the optimal steric and electrostatic match of A14 with the length of the pore. A14 is quasi-immobilized within the pore because is dimensionally and physico-chemically complementing the pore like a cork. This observation was then at the basis of our next step: instead of using immobilized DNA systems to dissect the sensing spots of the pore we used A14. Importantly, this gave us the opportunity to probe the pore using for the first time a free DNA system, which is much closer to the real translocation process of DNA than the streptavidin-based system.

When then using abase probes engineered on A14 we were able to recapitulate what we were already expecting and what was already apparent from the mere structure of the pore. When some of us solved its structure, we found out that aerolysin has 2 main constrictions at R220 and K238, which are much narrower (10 Å) than the averaged diameter of the pore (17 Å), and which also provide distinct electrostatic properties for DNA sensing. These 2 regions were already clear candidates to be the best sensing regions within the lumen. Our abase-A14 experiments were in fact able to recapitulate this expectation, validating both our initial hypothesis and the strategy to use A14-long oligonucleotides as quasi-immobilized probes for pore sensitivity analysis.

As for the concerns about different conformations, if this were the case, all abasic substitutions would have had similar behaviour, with similar duration. However, there is a strong difference depending on the abasic nucleotide positions on the sequence. And this difference matches perfectly the location of the two constriction points.

Therefore, as the results appeared to be internally consistent, we think that there is no need, for aerolysin pores at least, to use artificial immobilized DNA systems when a free DNA approach is able to deliver robust and consistent results. We are confident this more natural and less invasive protocol will be preferred by other investigators working on aerolysin or structurally similar pores to accurately probe pore properties. We have added a further discussion of these aspects on page 10 of the revised paper.

REVIEWERS' COMMENTS:

Reviewer #1 (Remarks to the Author):

The authors have adequately addressed all comments of this reviewers

Reviewer #2 (Remarks to the Author):

The purpose of this work is identifying the single nucleotide-sensitive spots in aerolysin. In my last comment, I have suggested experiments such as DNA immobilization, supposing that in these experiments the positions of each base in the pore for all DNA tested are identical, except the single abase that is placed at various positions in the pore to affect the conductance, and finally conclude the "sensing spot" locations.

In the revised paper, however, the authors did not conduct any similar or other conceivable experiments. The reasons given by the authors, such as that in the second paragraph "by using streptavidin the attached DNA have to be biotinylated at the 5'-side, and biotin can produce a larger flexibility of the whole system since its structure is more flexible than DNA. However, the main reason why other authors were using streptavidin to immobilize DNA is because they were working with proteins forming broader and shorter pores than aerolysin and, therefore, the DNA translocation was too fast to detect anything." is a guess. No evidence shows the negative effect of using this design in this pore.

Response to Reviewers

We thank the reviewers for their assessment of our manuscript and respond here below point-by-point to their concerns.

Reviewer 1

The authors have adequately addressed all comments of this reviewers

A1. Thanks so much!

Reviewer #2

The purpose of this work is identifying the single nucleotide-sensitive spots in aerolysin. In my last comment, I have suggested experiments such as DNA immobilization, supposing that in these experiments the positions of each base in the pore for all DNA tested are identical, except the single abase that is placed at various positions in the pore to affect the conductance, and finally conclude the "sensing spot" locations.

In the revised paper, however, the authors did not conduct any similar or other conceivable experiments. The reasons given by the authors, such as that in the second paragraph "by using streptavidin the attached DNA have to be biotinylated at the 5'-side, and biotin can produce a larger flexibility of the whole system since its structure is more flexible than DNA. However, the main reason why other authors were using streptavidin to immobilize DNA is because they were working with proteins forming broader and shorter pores than aerolysin and, therefore, the DNA translocation was too fast to detect anything." is a guess. No evidence shows the negative effect of using this design in this pore.

A2: Thanks for your comments. We have deleted the argument that the presence of biotin-streptavidin complex introduces larger flexibility into the system, focused on the ability to identify single point mutations and epigenetic marks without using a DNA threading protein.